# Surface darkening by abundant and diverse algae on an Antarctic ice cap

Alex Innes Thomson [1] ✉, Andrew Gray [2,3] ✉, Claudia Colesie [2], Naomi Thomas [1,4], Hannah Moulton [5], Peter Convey [5,6,7], Alison G. Smith [8], Peter Fretwell [5], Lloyd Peck [5] & Matthew P. Davey [1] ✉

Algal blooms play important roles in physical and biological processes on glacial surfaces. Despite this, their occurrence and impacts within an Antarctic context remain understudied. Here, we present evidence of the large-scale presence, diversity and bioalbedo effects of algal blooms on Antarctic ice cap systems based on fieldwork conducted on Robert Island (South Shetland Islands, Antarctica). Algal blooms are observed covering up to 2.7 km$^2$ (~20%) of the measured area of the Robert Island ice cap, with cell densities of up to $1.4 \times 10^6$ cells ml$^{-1}$. Spectral characterisation reveal that these blooms increase melting of the ice cap surface, contributing up to 2.4% of total melt under the observed conditions. Blooms are composed of typical cryoflora taxa, dominated by co-occurring Chlorophyceae, Trebouxiophyceae, and *Ancylonema*. However, morphological variation and genetic diversity in *Ancylonema* highlight the influence of regional endemism and point to a large and undercharacterised diversity in Antarctic cryoflora.

Antarctica's ice contains the single largest body of fresh water on the planet, which acts as the greatest surface contributor to planetary albedo and a significant driver of global air and oceanic circulation[1,2]. Recent warming trends in the Antarctic and globally, coupled with more frequent extreme heatwave events, are leading to dramatic changes in physical processes such as sea-ice formation and precipitation patterns across the region[3–6], as well as accompanying biological shifts[7,8]. Lying north-west of the Antarctic Peninsula, the South Shetland Islands are characterised by large ice caps, which cover up to 88% of the land mass across the archipelago[9]. Depending on the warming scenarios used, forecasting models have predicted potentially severe losses of snow and ice cover in the Antarctic Peninsula and Scotia Arc region (the maritime Antarctic) in approaching decades, with the near-complete disappearance of the ice caps on the South Shetland Islands by 2100[10]. The effects of such losses across the continent are likely to have far-reaching consequences, including

accelerated sea-level rise and feedback effects on the global energy balance[11,12]. This has led to concerted efforts to model and predict present and future ice-dynamics in Antarctica in order to improve our understanding of the drivers of loss and tipping points to habitat collapse, as well as to explore possible conservation and management measures and environmental futures for the region[10,13].

Biological albedo reduction (BAR) represents one contributing factor to mass loss from terrestrial cryosphere systems worldwide[14]. Polar and alpine glaciers are dynamic biological systems home to a diverse range of cryotolerant microorganisms, including cryoconite-forming bacteria and cyanobacteria, algae, fungi and higher meiofauna[15–17]. These communities are supported by photo-autotrophic production from cyanobacteria and algae inhabiting surface and near-surface snow and ice[18,19]. Blooms of algae have been shown to significantly influence the albedo and melt dynamics of the snow and ice systems they inhabit[14,20,21]. One group of algae in

[1]Scottish Association for Marine Science (SAMS), Oban, UK. [2]Global Change Institute, School of GeoSciences, University of Edinburgh, Edinburgh, UK. [3]Terrestrial Ecology Section, Norwegian Institute for Nature Research—NINA, Oslo, Norway. [4]Culture Collection of Algae and Protozoa (CCAP), Natural Environment Research Council, Oban, Argyll, Oban, UK. [5]British Antarctic Survey (BAS), Natural Environment Research Council, Cambridge, UK. [6]Department of Zoology, University of Johannesburg, Auckland Park, South Africa. [7]Millennium Institute Biodiversity of Antarctic and Sub-Antarctic Ecosystems, Santiago, Chile. [8]Department of Plant Sciences, University of Cambridge, Cambridge, UK. ✉e-mail: alexander.thomson@sams.ac.uk; andrew.gray@nina.no; matt.davey@sams.ac.uk

particular, the zygnematophyte algae *Ancylonema*[22], have been shown to contribute significantly to glacier and ice surface BAR in the Northern Hemisphere, largely through the accumulation of energy-absorbing dark purple phenolic pigments in their cells[23,24].

Algal blooms on Northern Hemisphere ice sheets have been shown to extend over >5000 km² and persist for up to 35 days or more[25]. On the West Greenland Sermersuaq (Ice Sheet), the accumulation of dense blooms of *Ancylonema* during the summer season has been estimated to increase melt rates of the ice surface by an average of 9–13%[20]. These melting rates were calculated using the spectral characteristics (herein referred to as hemispherical directional reflectance factors (HDRF)) of white, dirty or biologically darkened snow and ice to determine the instantaneous radiative forcing (IRF) of any algal cells present. In turn, the effects of biologically-driven glacial melt are predicted to further accelerate short and long-term feedback mechanisms in global albedo changes, climate warming and sea-level rise[20,26].

To date, algae blooms on glaciers have been widely reported across Northern Hemisphere glacial systems, but with only infrequent records in the Southern Hemisphere[27–29]. Unlike Northern Hemisphere and Arctic glacial systems, it remains unclear the extent to which algae occur on glacier and ice sheet surfaces in Antarctic regions, and if present, whether similar BAR processes occur. In Antarctica, terrestrial primary production is predominantly driven by cryptogamic communities such as mosses, lichens, and mats of algae and cyanobacteria[30–32]. Cryophilic algae are likely to play an important role in primary production and biogeochemical cycling across the Antarctic Peninsula region[18,33]. Yet, the contribution of glacier and ice sheet communities to these processes has been largely overlooked, highlighting a significant gap in our understanding of Antarctic ecosystems. Here, we provide a detailed report of algal blooms being present on an Antarctic ice cap (Robert Island, South Shetland Islands, maritime Antarctic, Fig. 1) and use remote sensing to provide quantitative estimations of their distribution and contribution to BAR and equivalent snow melt. In addition, we present data on the morphological and genetic diversity observed within the genus *Ancylonema* on Robert Island and discuss this in the context of Antarctic microbial endemism.

## Results

### Bloom density and extent on the Robert Island ice cap

The Robert Island ice cap had extensive blooms of algae across exposed glacial ice surfaces in January and February 2023 (Fig. 1). The habitat on top of the ice cap consisted of pillows of dry grey-purple to red névé-like weathering crust, interspersed by darker cryoconite and debris-laced ice hollows and water channels (Fig. S1). Within the 13.7 km² area analysed from the 6 February 2023 WorldView 2 satellite image, algae covered an estimated 2.7 km², almost 20% of the snow, weathering crust and glacial ice shown in Fig. 2. Blooms on the ice cap were observed from 7 January 2023 until the first significant late summer snowfall of the year that occurred on 24 February 2023.

Total cell counts (for all photosynthetic cells) showed an average of $2.5 \times 10^5$ ($\pm 2.7 \times 10^5$) cells ml⁻¹ of névé-snow or weathering crust, with a maximum count of $1.4 \times 10^6$ cells ml⁻¹. Specific counts of *Ancylonema* spp. made up ~10% of the total cell count from the main ice cap surface ($1.8 \times 10^4 \pm 2.9 \times 10^4$ ml⁻¹ of snow melt) ($n = 83$). The maximum recorded density of *Ancylonema* was $2.1 \times 10^5$ cells ml⁻¹, 23% of the cells in that sample.

### Field microscopy of glacier community samples

Light-microscopy of freshly sampled algae at the Robert Island field station and on samples exported to the UK showed a mixed algal

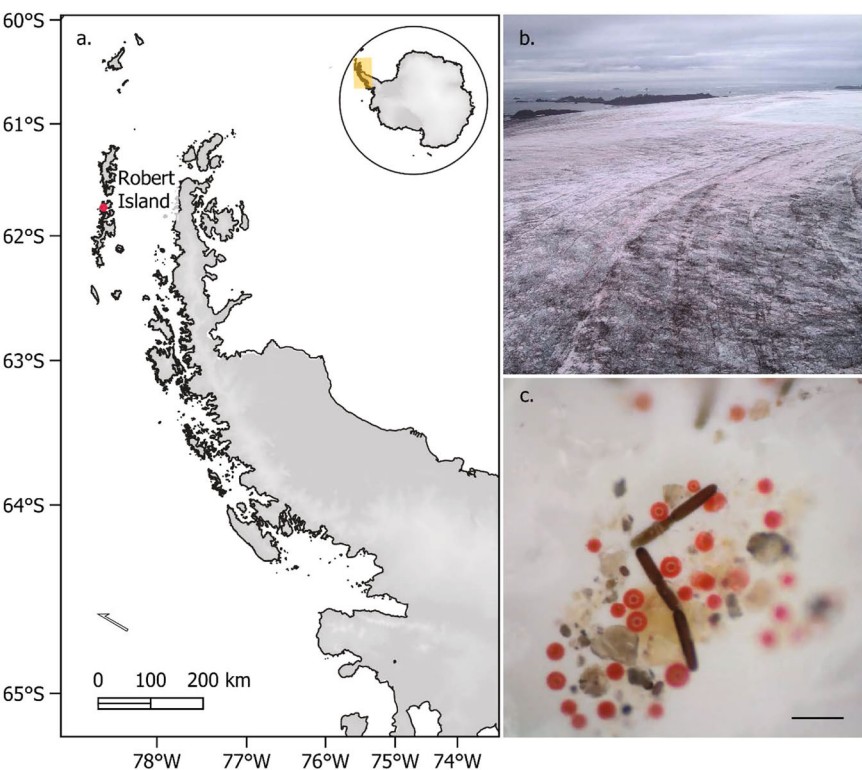

**Fig. 1 | The Robert Island ice cap.** Imagery of bloom extent and in-situ community composition from the austral summer 2022/23 field sampling campaign. **a** Location of Robert Island (South Shetland Islands, north-west of the Antarctic Peninsula in the maritime Antarctic). **b** Drone photograph showing red snow algae and purple glacier algae blooming on the Robert Island ice cap on 8 February 2023. **c** Dino-Lite micrography of the glacial ice surface algal community taken on weathering crust near the ice cap margin on 18 February 2023. Scale bar ~50 μm, based on the average large dark-purple *Ancylonema* cell dimensions.

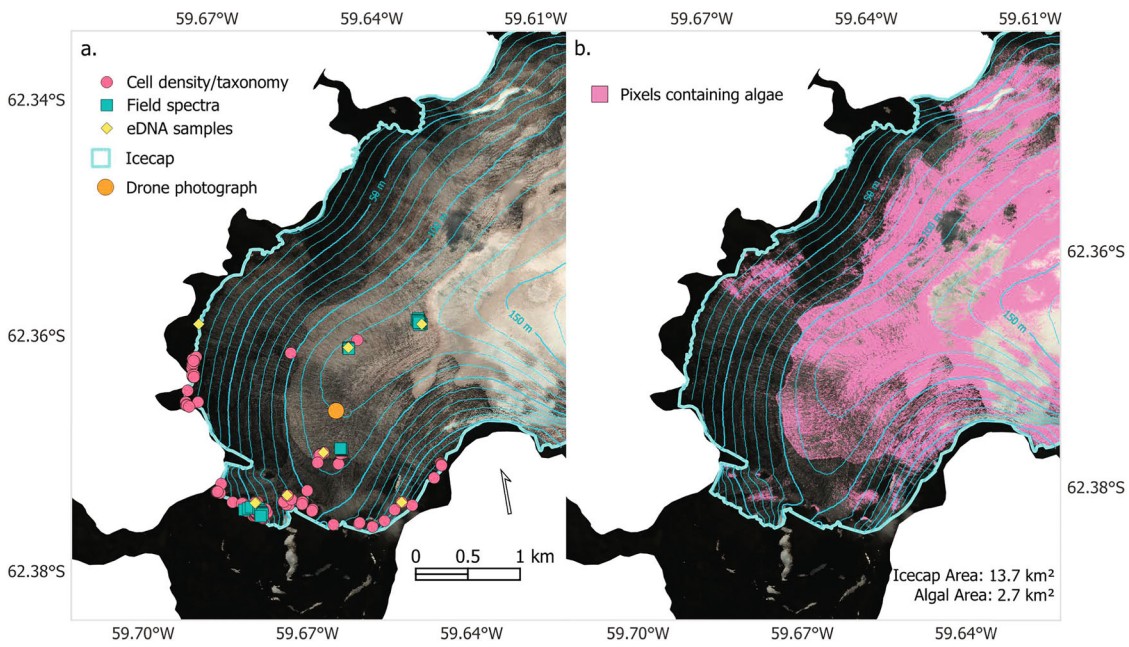

**Fig. 2 | WorldView 2 satellite image of the Robert Island ice cap and study area. a** Sampling locations on the Robert Island ice cap for algae cell counts (pink circles), field-spectra and paired cell density samples (green squares), community meta-barcode samples (yellow diamonds), and location of the drone photograph in

Fig. 1b. (orange circle). **b** Remote sensed algal extent on the Robert Island ice cap based on the WorldView satellite image from 6 February 2023 (pink pixel size 1.6 × 1.6 m). Contours are 10 m apart.

community across the ice cap consisting mainly of Chlorophyceae (mostly red *Sanguina*-type cells and occasional *Chlainomonas*-type cells), Trebouxiophyceae (*Raphidonema*-like cells), and Zygnematophyceae (*Ancylonema*), as well as occasional unidentified filamentous cyanobacteria (Figure. S2).

Of note was the wide diversity of cell morphologies observed amongst *Ancylonema* (Table 1, Fig. 3, Fig. S3). Cell types resembling the classic morphologies of the two species described from glacier habitats, *A. nordenskioeldii* and *A. alaskanum*[22], were observed across the ice cap. We also observed large morphotypes with strikingly different cell shapes (Table 1, Fig. 3b–d). Cell sizes amongst the larger morphotypes and smaller *A. alaskanum*-like morphotypes showed a noticeable deviation from previously published descriptions (Fig. 4a. and c). All cell types showed the characteristic purple-brown pigment accumulation of cryophilic *Ancylonema*, as well as typical longitudinal cell division and ice habitat requirement[22,34].

**Molecular diversity of glacier communities from Robert Island**
A metabarcode dataset was generated to characterise the eukaryotic community composition of the algae blooms occurring on the Robert Island ice cap. A pooled sample of eight locations from across the ice cap was sequenced across the 18S rRNA V4 region and processed into amplicon sequence variants (ASVs) in *dada2*[35]. 2.88 M reads were processed into 2595 ASVs, of which 352 were above the 0.01% confidence threshold (97.3% of total reads) (Table S1). The sequence data supported the microscopy observations, with Chlorophyceae and Trebouxiophyceae making up the largest proportions of reads, 26.1% and 20.4%, respectively (Fig. 5). This was followed by fungi (15.1%) and Charophyta (*Ancylonema*) (12.5%). *Sanguina*, *Ancylonema*, *Chlainomonas*, and number of ASVs pertaining to *Raphidonema*-like species made up six of the top ten most abundant ASVs (Fig. 5b). The fifth most abundant ASV, which was assigned as *Monochrysis sp.* (Chrysophyceae), was not immediately evident by microscopy in the field, however related taxa are known to be difficult to observe under light microscopy without specific equipment[36].

To further investigate the morphological diversity observed within *Ancylonema*, an ITS2 metabarcode dataset was generated for the same metabarcode sample pool to explore genetic diversity within the group. 18S and ITS2 ASVs were aligned against reference sequences for *A. nordenskioeldii* and *A. alaskanum* available from existing studies[22,37]. Sequences with >99% similarity against the reference 304 bp 18S V4 regions and >92% against the reference 347 bp ITS2 regions were assigned as putative haplotypes.

The pooled Robert Island sample library contained three 18S ASVs assigned as *Ancylonema*. 18S haplotypes H1 and H2 were 100% matches to the reference *A. nordenskioeldii* and *A. alaskanum* sequences, respectively. Haplotype H3 was situated basally to the two branches of *A. nordenskioeldii* and *A. alaskanum* but subsequent to the branch containing the recently described mesophilic *Ancylonema* species *A. palustre*[38] (Fig. S4). Within the 2.2 million ITS2 reads for the Robert Island pool, 20 haplotypes were found for *Ancylonema*, making up 17.6% of the reads and containing up to 29 bp variation across the 350 bp sequence alignment. Additional exploration of environmental sequence datasets from comparable glacier systems identified a further seven *Ancylonema* ITS2 haplotypes from locations in Greenland[39,40], Svalbard[40], and Alaska[40] (Tables S2 and S3). These were included in the phylogenetic analysis to provide a wider geographic comparison of *Ancylonema* phylogenetic diversity.

Sequence-structure alignment of ITS2 sequences, which can improve the phylogenetic resolution of primary sequences[41] and provide additional support for species delimitation via the presence of compensatory base pairs (CBCs)[42,43], broadly identified seven clades within the *Ancylonema* sequence pool, based on supported branches, and the presence of CBC positions between members of the assigned clades (Fig. 6). Two branches of the phylogeny, clades B and C, were specific to Robert Island. These contained the highest diversity and the most abundant ASVs from the Robert Island dataset, including H1, H2 and H4 (68.3% of total *Ancylonema* ITS2 reads). In comparison, clades D, E, and G supported sequences from Robert Island, as well as environmental sequences from Greenland, Alaska, and Svalbard, and reference sequences for *A. nordenskioeldii* and *A. alaskanum* from the

**Table 1 | Comparison of *Ancylonema* sp. morphotypes observed from the Robert Island ice cap (South Shetland Islands, maritime Antarctic)**

| Morphotype | Description | Average cell length (µm) | Average cell width (µm) | References |
|---|---|---|---|---|
| *RI-N* (Fig. 4a) | *A. nordenskioeldii*-like morphotypes. Singular, pairs, or in chains of up to 8 cells. (N = 121) | 48.6 (±9.9 µm) | 12.7 (±1.0 µm) | Procházková, et al. [22] |
| *RI-E* (Fig. 4b) | Elongated *A. nordenskioeldii*-like shape. Curved, asymmetrical form, tapering to a rounded point at cell ends. Generally singular or in pairs, with occasional chains of 4 or more cells. Similar morphotype reported from Tyndall Glacier, Chile. (N = 55) | 52.3 (±11.3 µm) | 9.6 (±1.0 µm) | Takeuchi and Kohshima, 2004[29] |
| *RI-T* (Fig. 4c, d) | 'Tapered' *Ancylonema* sp. morphotype. Asymmetric cell shape and distinctive pointed ends, occasionally 'tufted'. Singular or in pairs. Similar morphotype observed from Signy Island, South Orkney Islands, maritime Antarctic. (N = 54) | 41.3 (±9.3 µm) | 9.7 (±0.9 µm) | E. Broadwell (University of Bristol, UK) (pers. comm.) |
| *RI-A* (Fig. 4e-g) | *A. alaskanum*-like morphology. Large size range. Pill-shaped. Single or paired cells. (N = 219) | 14.3 (±4.6 µm) | 8.8 (±1.8 µm) | Procházková et al. [22] |
| *RI-C* (Fig. 4h) | Broad *A. alaskanum*-like morphology. Low cell length to cell width ratio. Single or paired cells. Similar large *M. berggrenii* morphotypes described from the Windmill Islands (Wilkes Land, continental Antarctic), though with reddish-brown pigment, as supposed to the dark purple-brown pigmentation observed in Robert Island samples (N = 45) | 24.5 (±4.8 µm) | 12.9 (±0.7 µm) | Ling and Seppelt[27] |

European Alps. A final group, Clade A, contained a low-frequency haplotype (H20) which aligned closely with environmental sequence data for a putative *Ancylonema* sp. identified by Remias et al.[37] This showed a distant basal relationship to the other *Ancylonema* clades and to the reference sequences for *A. nordenskioeldii* and *A. alaskanum*.

Analysis of sequence-structure alignments between representative haplotypes from each clade found one compensatory base change (CBC) between clades C and D and clade G, confirming previous analysis showing the presence of a CBC in helix I between *A. nordenskioeldii* and *A. alaskanum*[37]. Additional novel CBCs were identified in helix 3 between members of clade F and clades A, E, and G; between H18 (clade D) and clade G (*A. alaskanum*), and between H16 and all other clades. Multiple CBCs between Clade A and clades B-G confirmed previous analysis by Remias et al.[37], separating environmental sequence data for an unknown *Ancylonema* sp. clone from reference sequences for *A. nordenskioeldii* and *A. alaskanum*. Additional hemi-CBCs were identified between all clades in all four helices (Fig. S5, Table S4).

### Reflectance and instantaneous radiative forcing

Hemispherical directional reflectance factors (HDRF) of both bloom (red cell-dominated or areas containing *Ancylonema*) and non-bloom (snow or weathering crust with no detected cells) areas found on the Robert Island ice cap were obtained during the field season (Fig. 7a). We were unable to record in situ HDRF spectra of single taxonomic groups as all samples analysed were mixed communities of cells. All HDRF spectra from algae-dominated samples (Fig. 7a) showed absorption features associated with chlorophyll *a* at 680 nm[44], as well as carotenoid absorption between 400 and 550 nm[45]. The samples dominated by red *Sanguina*-like cells (Fig. 7a; dashed lines) had stronger absorption within the carotenoid signal region. Red cell-dominated samples had higher reflectance than *Ancylonema*-containing samples, independent of cell density (comparing dashed with solid lines in Fig. 7a), as they were predominantly recorded upon snow (Fig. 7a, blue line) rather than weathering crust (Fig. 7a, black line). However, the darker pigmentation of *Ancylonema* cells is also likely to have influenced this.

All sites that contained a measurable algal presence had lower broadband reflectance within the visible to near infra-red (VNIR) range (350 to 1000 nm), compared to clean weathering crust or snow (Fig. 7a). Absorption below 500 nm was also typically stronger for algal dominated patches compared with mineral debris, though broader quenching across the VNIR from heavy mineral loading within snow or ice often resulted in a lower broadband reflectance compared with the observed algal bloom densities.

There was a positive logarithmic relationship between instantaneous radiative forcing (IRF) and cell density (Fig. 7b) for both red cell-dominated and *Ancylonema*-containing samples. The relationship between cell density and IRF, shown in Fig. 7b, explained only 41% of variance within the data, indicating a significant contribution to IRF from secondary effects of cell presence, such as enhanced liquid water and mineral entrainment, as well as differences in snow and ice structure. Radiative forcing based on the mean recorded algal cell density would have contributed 0.45 to 2.36 % of total melt, depending on the amount of incoming solar radiation and daily temperatures (see Table 2 for a summary). Extrapolating over the area of bloom observed on the 6 February 2023, this would equate to $6.97 \times 10^6 \pm (5.36 \times 10^6)$ litres of meltwater derived from algal presence alone.

## Discussion

### Algal communities on ice caps and glaciers are an overlooked component of photoautotrophic systems in Antarctica

We have provided direct evidence of the presence and extent of photosynthetic ice cap communities in Antarctica. The densities and

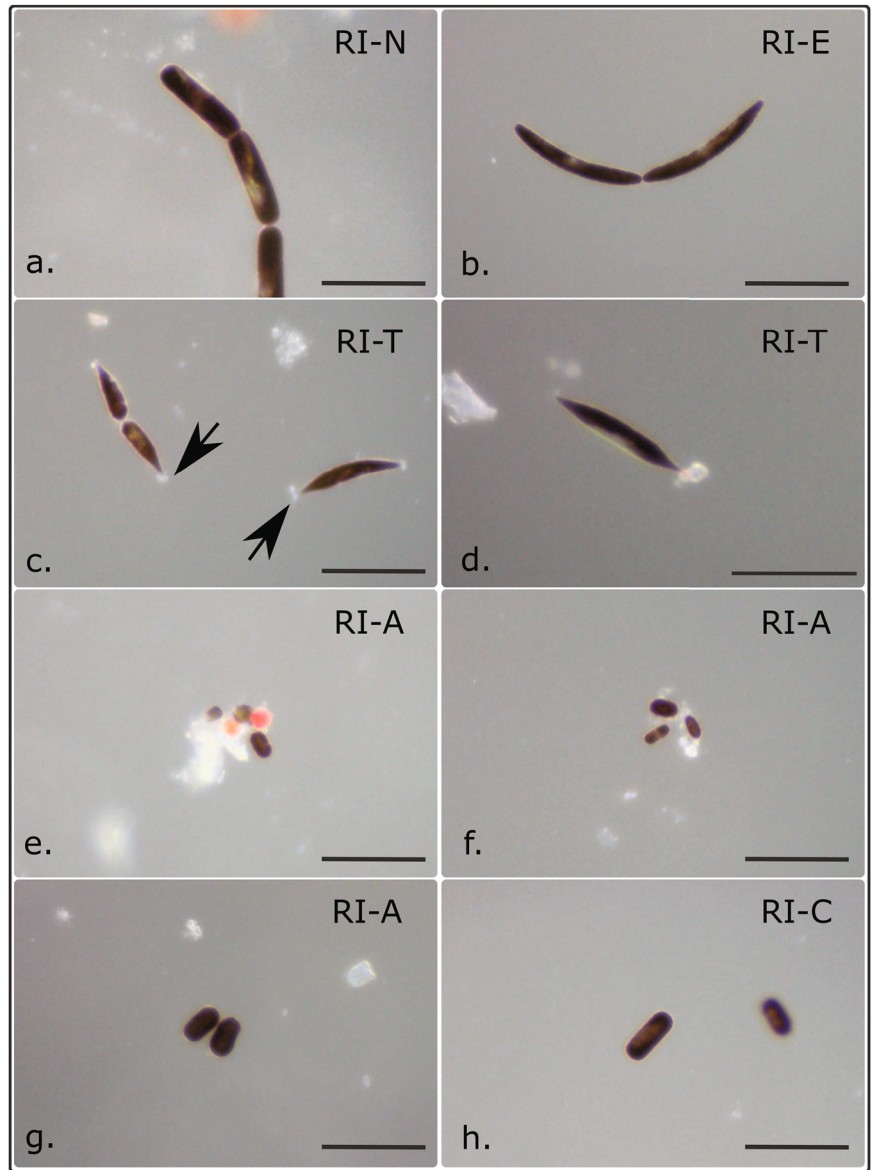

**Fig. 3 | Representative diversity of *Ancylonema* spp. morphotypes observed from Robert Island (South Shetland Islands, maritime Antarctic).** Micrographs of fresh samples imaged at the field station 1–5 h after collection. Scale bars 50 μm. **a** *Ancylonema nordenskioeldii* morphotype RI-N; **b** *Ancylonema* morph. RI-E; **c**, **d** cells of *Ancylonema* morph. RI-T, showing tufts or secretions at cell ends (arrows); **e**–**g** *A. alaskanum* morph. RI-A of various sizes; **h** the wider cell type of *Ancylonema* morph. RI-C. Representative of the images taken from 37 samples across the 198 ice cap sample locations. Higher-resolution micrographs are in Fig. S3.

scale of algal blooms on the Robert Island ice cap point to the significance of these communities as a major photoautotrophic system in the maritime Antarctic, distinct from other cryoflora assemblages, such as red and green snow algae from coastal snow packs[46–48]. The 2.7 km² estimate of bloom coverage on the Robert Island ice cap is equivalent to ~6% of the currently mapped area identified as terrestrial photosynthetic life in Antarctica[30]. The comparative extent of the Robert Island ice cap bloom and the suggested presence of similar biological communities on other ice caps in Antarctica[49] (Fig. S6), may make glacial ice and snow algal communities some of the most extensive (in terms of area) photoautotrophic systems within the Antarctic Peninsula and surrounding islands.

**Ice cap algae blooms on Robert Island differ in composition but not taxa from comparable communities on glaciers elsewhere**
Microscopy and metabarcoding analyses reveal that ice cap surface blooms on Robert Island consist of broadly similar cryoflora taxa compared to other glacier communities but differing proportional composition. The algal community on Robert Island is notably mixed, with a higher proportion of Chlorophyceae and Trebouxiophyceae than typically observed in Northern Hemisphere glacier communities. While *Raphidonema* has been widely reported on glacier and ice sheet surfaces, it is often found in low numbers[37,39,50,51]. The findings here align with previous data indicating a higher proportion of *Raphidonema* in Antarctic communities compared to those in the Northern Hemisphere[40]. Observations of bloom formation and morphological diversity within *Raphidonema*-like cell types from other locations in the maritime Antarctic (King George Island, Livingston Island, Signy Island and Cierva Point) also underscore the prevalence of this taxon in Antarctic cryoflora communities[28,52–54].

Mixed algal communities have also been documented on Northern Hemisphere glaciers[37,51]. However, many Northern Hemisphere glacial communities are dominated by *Ancylonema*[39,50,55], with Chlorophyceae typically becoming prominent only around or above the snow line. In contrast, on Robert Island, *Ancylonema* co-occurs with abundant Chlorophyceae and Trebouxiophyceae on the weathering

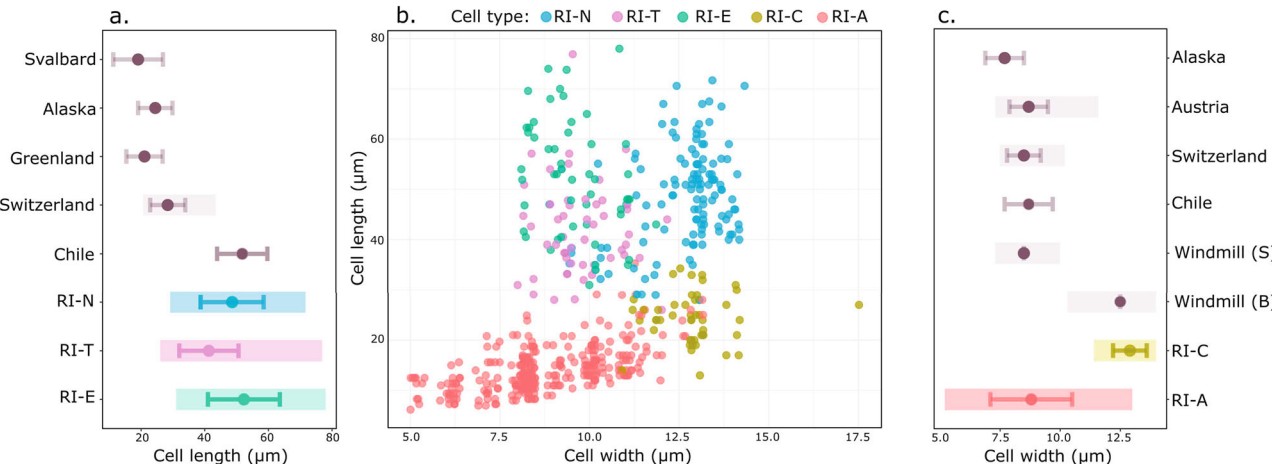

**Fig. 4 | Biometric comparison of *Ancylonema* spp. cells showing variation within and between Robert Island morphotypes and existing studies. a** Average cell length comparisons between large *Ancylonema* morphotypes observed from Robert Island and in the wider literature. **b** Cell length versus width distributions across all measured Robert Island *Ancylonema* morphotypes. A randomised jitter (offset) of ≤0.2 μm has been applied to each point along the cell width axis to aid visualisation. **c** Average cell width comparisons between smaller *A. alaskanum* type cells from Robert Island (including variant RI-C) and in the wider literature. Colours represent the different Robert Island morphotypes. Solid lines in (**a**) and (**c**) represent measurement mean (RI-N N = 121, RI-T = 54, RI-E = 55, RI-C = 45, RI-A = 219) and standard deviations, and shaded areas represent cell measurement ranges, where available. Average and SD measurements from the following locations and studies: Svalbard[34], Greenland[69], Alaska[50], Switzerland[22], Austria[22], Chile[29], Windmill Is. (S) (small cells)[27], Windmill Is. (B) (big cells)[27].

crust. Interestingly, *Ancylonema* cell counts from Robert Island were comparable to those reported from *Ancylonema*-dominated ice systems, despite constituting a smaller proportion of the total community (Table S5).

One possible explanation for the mixed community on Robert Island is the persistence of *Sanguina* cysts, which may have bloomed in overlying snow layers and have been deposited on the ice surface following snowmelt[37]. Alternatively, these results may suggest that a broader range of cryoflora taxa are adapted to glacier surfaces than previously recognised or that Antarctic lineages may exhibit particular adaptations to these environments. Further research will be needed to disentangle these factors, including temporal studies of microbial communities, enhanced in-situ monitoring of cell division and growth, and improved understanding of cryoflora physiology. Additionally, more detailed observations of the variations in snow and ice microhabitats could offer important insights into the environmental factors influencing community composition on glacier surfaces.

### Robert Island harbours a wide and unique diversity within *Ancylonema*

The morphological diversity of *Ancylonema* recorded from the Robert Island ice cap showcases a previously under-explored diversity of this group in Antarctica, one that is distinct from previous observations in the Northern Hemisphere and one that provides a potential example of regional-endemism in cryoflora communities in Antarctica. Morphological diversity amongst *Ancylonema* spp. has been observed elsewhere in the Southern Hemisphere, including from the Windmill Islands (Wilkes Land, continental Antarctic)[27], Tyndall Glacier (Patagonia)[29], King George Island (South Shetland Islands, maritime Antarctic)[28], and Signy Island (South Orkney Islands, maritime Antarctic) (E. Broadwell, University of Bristol, UK (pers. comm.)). The consistent differences observed versus Northern Hemisphere *Ancylonema* morphotypes support the presence of a unique diversity of these glacier-inhabiting algae in the Southern Hemisphere.

Metabarcoding data provided additional evidence for a large and distinct diversity within *Ancylonema* on Robert Island. A total of three 18S ASVs and 20 ITS2 ASVs were identified as *Ancylonema* haplotypes. In comparison, a recent analysis of combined Northern Hemisphere samples from 18 locations across Greenland, Svalbard, Sweden, Switzerland and Austria reported a total of five 18S ASVs and 16 ITS2 ASVs

for *Ancylonema* spp. across the same marker regions[37]. Inclusion of environmental sequences from available Northern Hemisphere datasets in our ITS2 phylogeny, as well as reference sequences for the currently described species of *Ancylonema*, revealed closely related haplotypes present on Robert Island, as well as distinct 'Antarctic' haplotype groups, with no representation from Northern Hemisphere datasets.

This evidence for the bi-polar presence of the currently reported species of glacier *Ancylonema* suggests a broad cosmopolitan biogeography of certain lineages of *Ancylonema* in glacial habitats worldwide. In addition, the observed morphological and ITS2 variation from Robert Island suggests the presence of a distinct Antarctic or Southern Hemisphere *Ancylonema* diversity. The distribution of Robert Island haplotypes amongst regional-endemic and cosmopolitan clades aligns with other examples of Antarctic microbial endemism[56,57] and raises intriguing questions about biogeographic patterns, refugia, and divergence histories in cryophilic *Ancylonema*. Whether Antarctica represents a 'cryo-refugia' for ice-dependant lineages of microalgae and whether additional diversity and regional variation remain to be discovered amongst Antarctic cryoflora, will depend on further sampling and analysis of glacial communities from elsewhere in Antarctica and the Southern Hemisphere.

### Algae blooms contribute to melting of the Robert Island ice cap
Our findings highlight the contribution of cryoflora communities to BAR on the Robert Island ice cap. There are still relatively few studies calculating BAR from glacier ice surfaces, in part due to difficulties in unpicking organic, inorganic, and weathering crust morphology contributions to IRF[24,58]. Cook et al.[20] reported average cell counts of $2.9 \times 10^4$ cells ml$^{-1}$ ($\pm 2.0 \times 10^4$) for high biomass areas of *Ancylonema* dominated blooms on the Greenland Sermersuaq. This was comparable to the average *Ancylonema* spp. cell counts from the Robert Island ice cap ($1.8 \times 10^4$ cells ml$^{-1}$ ($\pm 2.9 \times 10^4$)), and less than the average total algal cell count ($2.5 \times 10^5$ cells ml$^{-1}$ ($\pm 2.7 \times 10^5$)). However, biologically enhanced melt rates for high biomass algae-containing surfaces on the Greenland Sermersuaq were greater than those calculated for average cell densities on Robert Island. Based on average solar radiation conditions on Robert Island between 4 January and 16 February 2023, we recorded a daily mean IRF of 16.6 W m$^{-2}$, similar to values reported for Antarctic red snow algae[46] but much lower than the 116 W m$^{-2}$ reported

**a**

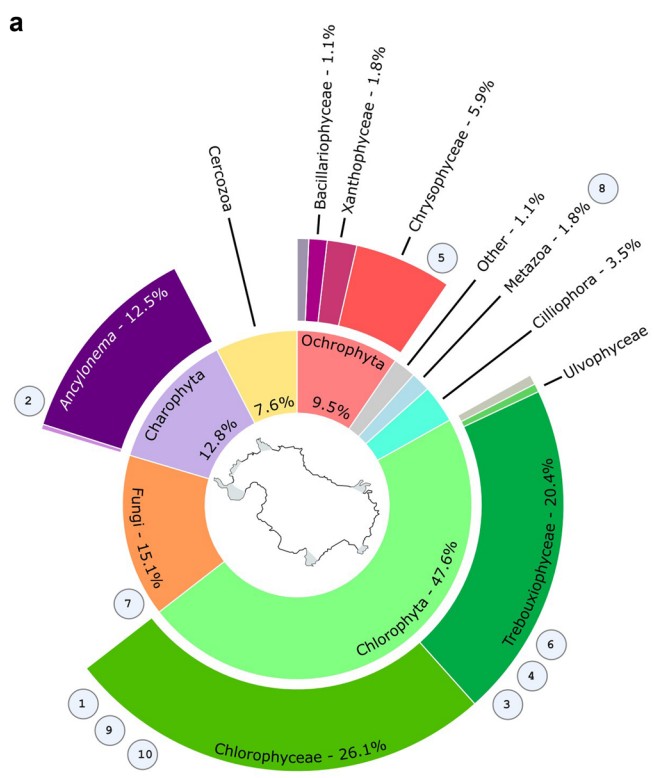

**b**

| | Blast-n ID | Accession | Read % |
|---|---|---|---|
| ① | *Sanguina* sp. | OR224454.1 | 11.0% |
| ② | *Ancylonema nordenskioeldii* | MW922838.1 | 9.9% |
| ③ | *Koliellopsis inundata* | MT274431.1 | 9.3% |
| ④ | *Raphidonema nivale* | AB488604.1 | 6.0% |
| ⑤ | *Monochrysis* sp. | AB058364.1 | 3.4% |
| ⑥ | *Raphidonema nivale* | AB488604.1 | 3.3% |
| ⑦ | Unknown Ascomycete | n/a | 2.5% |
| ⑧ | *Bdelloidea* sp. | LC128703.1 | 1.8% |
| ⑨ | *Chlainomonas* sp. | OP003985.1 | 1.8% |
| ⑩ | *Hammakko caudatus* | AB451188.1 | 1.8% |

**Fig. 5 | Biological community overview of the Robert Island ice cap (South Shetland Islands, maritime Antarctic) from 18S metabarcoding. a** 18S V4 metabarcode community composition from the Robert Island ice cap sequencing pool. Circled numbers refer to the top ten ASVs as outlined below. **b** The top 10 ASVs and their proportion within the 18S V4 dataset and the nearest NCBI blast-n accession (>99.6% similarity).

for high biomass surfaces on the Greenland Sermersuaq[20]. This difference was mostly attributed to the cloudier conditions typical of Robert Island. These estimates suggest that algae-enhanced melt rates on the Robert Island ice cap (~ 0.5 cm day$^{-1}$; or 0.5–2.4% of total melt) are less than those observed on the Greenland Sermersuaq (1.35 ± 0.01 cm day$^{-1}$; or 9.0–13.0% of total melt)$^{(20)}$, but still represent a significant contribution of BAR to surface mass balance processes on the Robert Island ice cap.

Our calculations from the area of the study (shown in the World-View satellite image in Fig. 2) are likely an underestimate of the true coverage of algae, as the methodology used was based on the detection of algae from snow and may be confounded by heavy mineral debris on the glacier surface which can obscure the chlorophyll signal within a pixel[59]. In addition, since we were unable to sample areas of dominant or pure *Ancylonema*, there is also uncertainty regarding the effectiveness of the method in detecting *Ancylonema* dominated blooms on weathering crust and glacial ice. Developing methods to distinguish the albedo contributions of *Ancylonema* and other biological communities from inorganic debris and weathering crust

morphology remains a significant challenge. Addressing this is likely to require further point-sampling and drone-based ground-truthing across a diverse range of glacier surfaces and regions.

The extent of our mapping was limited to the area of the study. However, anecdotal observations from Sentinel-2 imagery (European Space Agency) suggest that the blooms in the austral summer 2023 covered a much larger area of the Robert Island ice cap than studied here. Similar dark and red pigmented ice fields have been observed on other Antarctic ice caps, including Nelson Island (South Shetland Islands) and Signy Island (South Orkney Islands) (WorldView imagery −Fig. S6), and in recent drone-based studies on Livingston Island (South Shetland Islands)[49]. These findings point to the potentially widespread presence of algal communities on glaciers and ice caps in the maritime Antarctic and highlight the need for wider ground-truthing and study of Antarctic cryosphere systems to determine the distribution and extent of these blooms, assess their impact upon melt rates, and evaluate their contribution to regional terrestrial productivity.

The findings presented here provide important evidence of the contribution of glacial communities to cryosphere biodiversity, photoautotrophic community presence, and ice surface darkening in Antarctica. The community composition of the Robert Island ice cap reflects the global diversity of cryoflora but highlights the critical role of biogeography and regional endemism in shaping the distinct characteristics of cryosphere regions. Mapping data from Robert Island, combined with evidence from comparable systems and studies[20], demonstrates the potential scale of algal blooms across Antarctica's ice caps and its ramifications. The combined evidence streams indicate the presence of a major photoautotrophic ecosystem harbouring a unique biodiversity on Antarctica's terrestrial ice caps and point to significant biological contributions to glacial melt dynamics in the maritime Antarctic.

## Methods
### Field sampling on Robert Island
Ice cap surface samples (Fig. 2) were collected in 50 ml sterile plastic sample tubes at 198 locations across the northwest part of Robert Island's ice cap (South Shetland Islands, maritime Antarctic; 62°22'44.4"S 59°42'1.78"W) between 7 January 2023 and 18 February 2023. Sampling locations were chosen to encompass a range of snow and ice surface types.

Once a patch was identified, spectral reflectance, the ratio of upwelling radiance to downwelling irradiance, was recorded in triplicate for each sample area using a Spectral Evolution PSR 3500+ field spectrometer equipped with 4° field of view (FOV) foreoptics. Reflectance was measured across the 350 to 2500 nm wavelength range, following the bi-conical methodology by Gray et al. [33] using a calibrated, 98% Spectralon panel as an irradiance reference, bracketing each measurement taken.

50 ml and 15 ml samples were then taken from the top 3 cm of snow or weathering crust within the FOV of the spectrometer, avoiding compaction of snow or weathering crust in the tube. Samples were transported within 1–3 h to the laboratory at the Chilean (INACH) station Luis Risopatrón, Coppermine Cove, on Robert Island for sample processing, cell counts and field microscopy.

### Sample processing and microscopy
Sample tubes were thawed at air temperature (< 10 °C) within the research station within 1–5 h of sampling. Cell counts were obtained by counting auto-fluorescent cells in 2 × 10 μl aliquots from each sample on an Invitrogen Countess 3 FL Automated Cell Counter. *Ancylonema* cell counts were obtained manually by identifying *Ancylonema* cells from the Countess 3 FL output images. Due to the poor resolution of the Countess images it was not possible to calculate individual cell counts of other species groups present. An additional 10 μl aliquot was

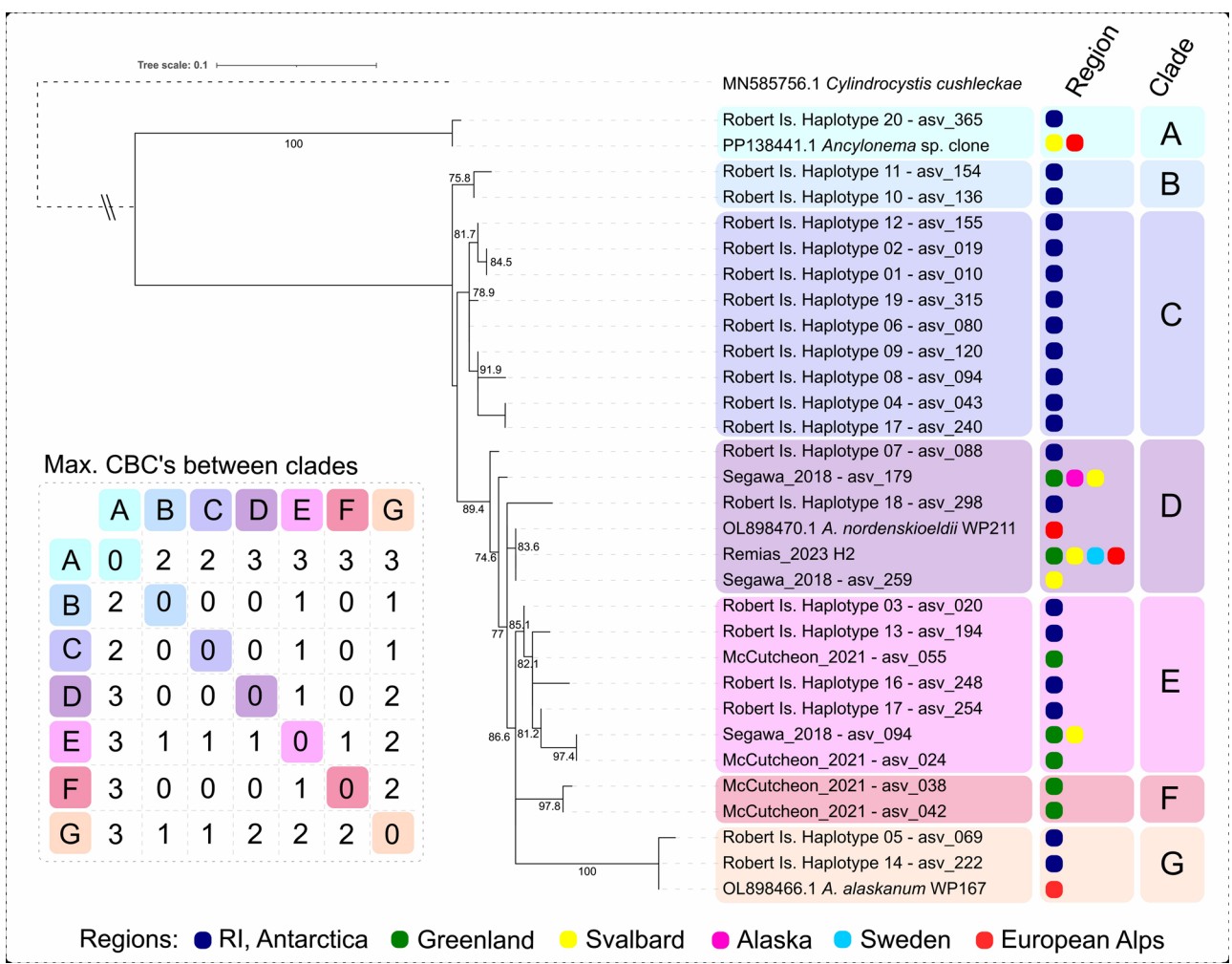

**Fig. 6 | *Ancylonema* spp. ITS2 haplotype diversity and phylogenetic placement from Robert Island and wider studies.** Sequence-structure phylogenetic maximum-likelihood tree of *Ancylonema* ITS2 haplotypes from Robert Island and elsewhere, including reference and environmental sequences for *A. nordenskioeldii*, *A. alaskanum*, and *Ancylonema* sp.[22,37], environmental sequence haplotypes from Greenland, Svalbard, Alaska, Sweden, and the European Alps[37,39,40], and an outgroup reference, *C. cushlekae*. Sequence data for "Remias_2023 H2" are in supplementary data for Remias et al. (2023)[37]. Haplotype occurrence by region is indicated in the 'Region' column. Maximum CBC positions between members of assigned clades are indicated in the inset table. Read distributions for ITS haplotypes are in Table S2.

taken for field microscopy and imaging of the fresh samples using a Dino-lite microscope (Taiwan). Biometric measurements of cell length and width for *Ancylonema* cells were taken from calibrated Dino-lite imagery (*n* = 570 cells from 37 samples).

Samples for additional microscopy in the UK were prepared by centrifuging a 15 ml aliquot of the sample tube for 2 min at 1100 × *g*, removing the snow-melt supernatant, resuspending the pellet, and transferring it to a 2 ml microfuge tube before being stored at −20 °C. Samples were collected from Robert Island by the RRS *Sir David Attenborough* (SDA) on 24 February 2023 where they were stored at −20 °C and transported to the UK.

**Molecular data preparation and analysis**

Samples for metabarcode community analysis were collected alongside spectral and biological samples from across the Robert Island ice cap. Cell pellets from 15 ml snow samples were preserved in Zymo DNA/RNA shield and stored, similarly to microscopy samples, for transport to the UK. Community genomic DNA was extracted using Qiagen Plant DNEasy kits with the addition of a PowerSoil bead-tube lysis step. The 18S rRNA V4 region was targeted using eukaryotic primers 528_F (5′ GCGGTAATTCCAGCTCCAA) and 706 R (5′ AATCCRAGAATTTCACCTCT)[60]. The ITS2 region was targeted using

forward primer 5.8SbF (5′ GATGAAGAACGCAGCG)[61] and a bespoke reverse primer ITS4_RI_R (5′ TTTCTTTTCCTCCGC), adapted from primer ITS4R[62] to better capture and amplify Zygnematophyceae, as well as Chlorophyta, sequence diversity. PCR amplification was undertaken in triplicate for individual sampling stations across Robert Island, using NEB Q5 polymerase (New England Biolabs), and 25 cycles of amplification at 98 °C for 10 s, 57 °C for 20 s and 72 °C for 20 s. Eight stations from across the ice cap were chosen based on geographic coverage (Fig. 2) and pooled at equimolar concentrations for downstream library preparation. The final library was prepared and sequenced on a NovoSeq (250 bp PE) by Novogene UK (Cambridge, UK).

Raw sequences were trimmed of adaptor and primer sequences and separated into 18S and ITS2 reads using Cutadapt v4[63]. Paired-end reads were then quality-filtered, denoised, and merged into ASVs using *dada2*[35]. ASVs at >0.01% across the total read count for 18S were retained for community analysis. Community-wide species inference for 18S rRNA was analysed against the *Silva SSU 132* database in *dada2*. Taxonomic assignments for the top 100 most abundant ASVs were additionally validated by aligning each ASV against the NCBI *blast*-n database and comparing outputs.

ASVs relating to *Ancylonema* spp. were identified by aligning reference sequences for *A. nordenskioeldii* and *A. alaskanum* against

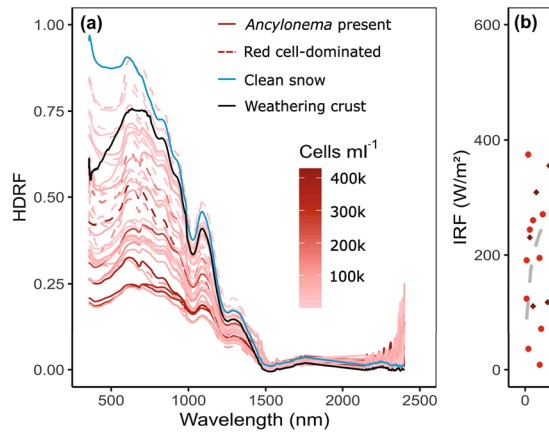
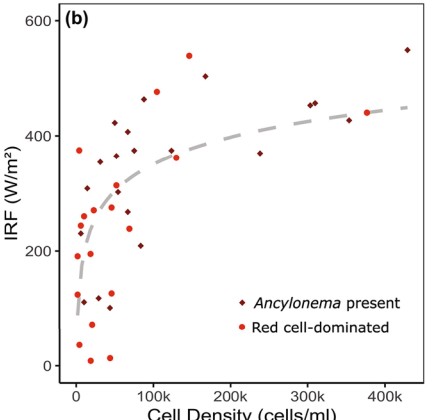

**Fig. 7 | Field spectral data from the Robert Island ice cap (South Shetland Islands, maritime Antarctic). a** Hemispherical directional reflectance factors (HDRF) for cryoflora on the Robert Island ice cap. Spectra with *Ancylonema* present are shown with solid red line, red cell dominated spectra shown with red dashed line, clean snow with no visible algae or dust with solid blue line, weathering snow crust with solid black line. Colour scale shows cell density (cells ml⁻¹).
**b** Instantaneous radiative forcing (IRF W/m²) compared with cell density (cells/ml) for the samples with *Ancylonema* present (diamond) and red pigment-producing cryoflora (circle) observed on the Robert Island ice cap.

**Table 2 | Melt and algal contribution to melt, modelled using PAR and temperature data from the 1 February and 6 February 2023**

| Date | Surface | Average air temperature (°C) | Average PAR ($\mu$mol m$^{-2}$ s$^{-1}$) | Melt from algae (mm) ± (s.d) | Total Melt (mm) | Percentage Contribution |
|---|---|---|---|---|---|---|
| 01/02/23 | Snow | 3.15 | 167 | 0.08 ± (0.07) | 18.4 | 0.46 |
| 01/02/23 | Weathering crust | 3.15 | 167 | 0.21 ± (0.16) | 21.7 | 0.98 |
| 06/02/23 | Snow | 2.42 | 399 | 0.20 ± (0.16) | 11.6 | 1.74 |
| 06/02/23 | Weathering crust | 2.42 | 399 | 0.51 ± (0.39) | 21.4 | 2.36 |

the ASV fasta output files. ASV sequences were identified as positive matches if sequence identity was >99% for the 18S rRNA region (~306 bp and up to 3 bp variation) and >92% for the ITS2 region (~340 bp and up to 28 bp variation). Positive ASV matches at >0.05% presence across the total read count for each marker were kept for further analysis.

Additional environmental ITS2 sequences for *Ancylonema* were compiled from available studies from glacier systems worldwide for inclusion in the phylogenetic analysis (Table S2). Datasets from published studies that targeted the ITS2 region were first screened for *Ancylonema* presence using the Sequence Read Archive Nucleotide BLAST function. SRA accessions with >100 *Ancylonema* reads were downloaded and processed in *dada2* in the same manner as the Robert Island dataset. The same threshold read coverage of > 0.05% that was applied to the Robert Island data set was applied to all *Ancylonema* haplotypes from the added environmental datasets.

18S rRNA phylogenetic placement was investigated for *Ancylonema* matches from the Robert Island library by aligning these against known reference sequences for *A. nordenskioeldii and A. alaskanum*, as well as a wider selection of Zygnematophyceae reference sequences, adapted from a previous phylogenetic analysis[22,38]. Maximum likelihood phylogenetic trees for 18S rRNA were constructed in the *IQtree* web server[64] based on the best-fit model parameters identified for each target region in *MEGA-11*[65]. Support for phylogenies was assessed based on ultrafast bootstrapping at 1000 iterations.

Interspecific and intraspecific genetic diversity was investigated within ITS2 haplotypes identified as *Ancylonema* spp. using a sequence-structure analysis framework. Secondary structure analysis has been suggested to improve phylogenetic resolution obtained from primary sequences, in particular across the ITS2 region[41]. In addition, specific structural features can be analysed for the presence of compensatory base changes (CBCs), which can provide additional support for species delimitation[42,43].

For the Robert Island and additional environmental sequence datasets, rRNA regions of ITS2 for each *Ancylonema* haplotype were first identified using the ITS2 database web server (http://its2.bioapps.biozentrum.uni-wuerzburg.de/). rRNA sequence structure was predicted using the single sequence - maximum expected accuracy function in *RNAStructure* V6.4[66]. Folded ITS2 rRNA models were inspected visually in *StructureEditor* v1.0 for the presence of core motifs expected in Chlorophyta: four helices, a U–U mismatch in helix II, and a triple A motif between helix II and III[42]. Where necessary, manual editing of the fold structure was undertaken in *StructureEditor*, using previously published *Ancylonema* spp. models as a reference[37]. The edited sequence-structure was exported as a .xfasta file and imported into *4SALE*[67] for sequence-structure alignment and CBC analysis. Sequence-structure alignments using *CLUSTAL*[68] were manually checked and edited to best reflect the maximum expected accuracy fold models provided by *RNAStructure*. CBC analysis was performed in *4SALE* on the edited sequence-structure alignment. A one-letter alignment was exported from *4SALE* and imported into *IQtree* for the construction of maximum-likelihood phylogenies. Additional ITS2 reference sequences for *A. nordenskioeldii* (OL898470.1), *A. alaskanum* (OL898466.1), and an environmental sequence for *Ancylonema* sp. (PP138441.1) were also included in the analysis, as well as a close outgroup reference sequence for *Cylindrocystis cushleckae* (MN585756.1). Support for the consensus tree was assessed based on ultrafast bootstrapping at 1000 iterations[69].

**Spectral reflectance and instantaneous radiative forcing**
Mean absolute spectral reflectance for each sample location was calculated as hemispherical directional reflectance factors (HDRF) using Eq. 1. After microscope and cell count analysis of the snow/weathering

crust sample retrieved from the spectrometer's field of view, we were able to classify the spectra into communities dominated by *Sanguina*–like red cells ($n = 19$) and mixed communities containing *Ancylonema*-like purple cells ($n = 22$). We also recorded spectral reflectance of clean (containing no fluorescing cells) glacial weathering crust ($n = 5$) and clean snow ($n = 5$).

$$\rho_\lambda = \frac{L_{e\lambda(Target)}}{L_{e\lambda(Panel)}} \beta_\lambda \qquad (1)$$

Where $\rho$ is spectral reflectance, $L_{e\,\lambda(Target)}$ is the radiance of the target surface, $L_{e\,\lambda(Panel)}$ is the mean radiance of the Spectralon panel, taken before and after the target measurement, and $\beta_\lambda$ is the calibrated reflectance of the Spectralon panel.

As we did not have a cosine collector for the spectrometer, we were unable to calculate albedo with the spectral reflectance data, instead, we calculated instantaneous radiative forcing (IRF) using our HDRF following Ganey et al. [26] and Khan et al. [46] using Eq. 2.

$$IRF = \int_{400}^{700} \left( E_e(\lambda) \left( \rho_{clean(\lambda)} - \rho_{algae(\lambda)} \right) \right) \Delta\lambda \qquad (2)$$

Where $E_e$ is the irradiance flux density (W m$^{-2}$), derived from a Hobo S-LIA-M003 PAR sensor, logging close to the ice cap in 5 min intervals, 2 m above the surface throughout our fieldwork. $\rho_{(clean)}$ is the spectral reflectance of clean snow or ice and $\rho_{(algae)}$ is the spectral reflectance of each algal patch sampled, each integrated over 400–700 nm. IRF was compared to measured cell density using the median value of PAR experienced during daylight hours whilst in the field (351 μmol m$^{-2}$ s$^{-1}$). Analysis was performed using R Studio (version 2022.02.3).

### Remote sensing and calculation of biological contribution to melt

The extent of algae on the Robert Island ice cap was determined within a WorldView 2 satellite image (Maxar Technologies) taken on 6 February 2023, coincident with our fieldwork. Using the methodology outlined in Gray et al. [59], algal presence on the ice cap was classified based on chlorophyll-*a* absorbance within each pixel. This was calculated on a per-pixel basis using Eq. 3, which describes the scaled integral of WorldView 2's Band 4 versus Band 3 and Band 5, the centres of which are located within and on either shoulder of the chlorophyll absorption feature.

$$I_{B5} = \frac{R_{B4}(\lambda_{B6} - \lambda_{B5}) + R_{B6}(\lambda_{B5} - \lambda_{B4})}{\lambda_{B6} - \lambda_{B4}} - R_{B5} \Bigg/ \frac{R_{B4}(\lambda_{B6} - \lambda_{B5}) + R_{B6}(\lambda_{B5} - \lambda_{B4})}{\lambda_{B6} - \lambda_{B4}} \qquad (3)$$

Where $I_{B5}$ is the scaled integral of Band 5, $R_{Bn}$ is the HDRF for Band "$n$", and $\lambda_{Bn}$ is the wavelength at the centre point of Band "$n$". Pixels were classified as containing algae if they had a $I_{B5} > 0$[60].

Though this method was originally optimised for detecting algae in snow, it successfully separated every HDRF sample from the background snow or weathering crust when Eq. 3 was applied to the collected HDRF spectra. However, an ice cap surface is more complex than a pure snow environment, with snow, weathering crust, glacial ice and different loadings of mineral debris. We do not have enough field spectra from this study to fully deconvolve cell densities based on chlorophyll-*a* absorption alone. Correlation between Eq. 3 derived and measured cell densities for the HDRF samples we collected only had an $R^2$ of 0.24. We therefore report a binary presence/absence classification of algal cells within each pixel, where an algae-containing pixel has a value of greater than 5452 cells ml$^{-1}$ (the y-intercept of the integral versus cell density reported in Gray et al. [59]), rather than making a cell density estimate from the WorldView image. The ice cap was manually digitised within the image and only the algae present upon the ice cap

included for analysis. Bloom extent was calculated based on the sum of each pixel classified as containing algae.

Bloom contribution to melt was calculated based on the cell density/IRF relationship in Fig. 5b (Eq. 4) and using the average recorded total cell density ($2.5 \times 10^5$ cells ml$^{-1}$). A simple numerical energy balance model was run in R (Rstudio Version 2024.04.2) with calculations based on snow and weathering crust surfaces. Albedo and density of each surface were set using average in situ broadband HDRF and the recorded density of each surface. We used weather station data (HOBO U30) in 5 min intervals from two contrasting days: 1 February 2023 where the average temperature was 3.15 °C and PAR was 167 μmol m$^{-2}$ s$^{-1}$ and 6 February 2023 (the date of the WorldView image) where the average temperature was 2.42 °C and PAR was 399 μmol m$^{-2}$ s$^{-1}$ (Data available at https://doi.org/10.5281/zenodo.14106494). IRF is dependent on the amount of solar radiation incident on the snow/ice surface and so comparing between sunny and cloudy conditions gave a good insight into the variability of algal influence upon melt.

$$IRF(Wm^2) = 5.52 \ln(celldensity) - 24.03 \qquad (4)$$

The energy balance model used both radiative and sensible heat flux contributions. Sensible heat flux was calculated using air temperature, wind speed, and a heat transfer coefficient, which was adjusted according to relative humidity for each 5-minute interval. The incoming solar radiation (PAR) was converted to energy and adjusted by the surface albedo and algal radiative forcing for each of the two surface types. For each time step, meltwater equivalent was calculated based on the total energy contribution of solar and sensible heat components. Daily melt volumes were computed by summing the 5 min interval results, providing estimates for both algal-induced melt and melt without algae present. Error was calculated based on one standard deviation from the IRF/cell density relationship. Melt water equivalent was extrapolated over the area of observed bloom, with error derived from IRF/cell density and the standard deviation of cell densities observed within our samples. Full code for these calculations is available at https://doi.org/10.5281/zenodo.14106494.

### Reporting summary

Further information on research design is available in the Nature Portfolio Reporting Summary linked to this article.

## Data availability

Field sample location data, cell counts, morphology metrics and remote sensing output (red snow algae vector for Robert Island) are available at the NERC EDS UK Polar Data Centre: https://doi.org/10.5285/afd8a811-6e5e-4441-a409-0e6924c36fc4. Environmental sequence data are available at the NCBI Sequence Read Archive (SRA) BioProject: PRJNA1158721 https://dataview.ncbi.nlm.nih.gov/object/PRJNA1158721. Temperature and PAR meteorological station data for remote sensing modelling are available at https://doi.org/10.5281/zenodo.14106494.

## Code availability

Environmental sequence data were processed using the publicly available R package *dada2* v1.26.0 in R v 4.2.1 "funny-looking-kid". Phylogenetic analysis was undertaken using the IQTree web server and IQ-TREE multicore version 1.6.12. Instantaneous radiative forcing calculations were performed in R Studio version 2022.02.3. Remote sensing analysis was undertaken in QGIS. Code for melt calculations is available at https://doi.org/10.5281/zenodo.14106494. All code can be made available upon request.

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

## Acknowledgements

This research was funded under joint UKRI NERC grants NE/V000764/1 and NE/V000896/1 awarded to M.P.D., A.G., C.C., P.C., P.F., L.P., A.G.S. Work was also carried out under Norwegian Romsenter grant 74CO2406 (AG). We are especially grateful to the Chilean Antarctic Institute (INACH) for their logistical support for the Robert Island fieldwork in 2022/23, in particular Rene Quinan, Diego Haeger, Ignatio Reyes, Bruno Escare, Juan Bravo and Andrés López Lara, as well as Charlotte Walshaw for her teamwork. We also thank the staff and crew of the BAS Rothera Research Station, Antarctica and the RRS *Sir David Attenborough*, staff at the King George Island Escudero Base (INACH) and all logistics and support staff at BAS. We thank the NERC Field Spectroscopy Facility (NERC FSF), Edinburgh, for the loan of the ground and drone-based spectroscopy equipment (loan reference 858.1120). Open Access fees for this article are funded from the UKRI Open Access Block Grant 2024/25. All samples were exported from Antarctica under licence and imported into the United Kingdom under Articles 8 and 48(1) of Regulation (EU) 2016/2031 and Delegated Regulation (EU) 2019/829 and plant health authorisation no PH/2/2023.

## Author contributions

A.G., A.G.S., P.C., P.F., L.P., C.C., and M.P.D. conceived the project and obtained funding. A.I.T., A.G., P.C., C.C., and M.P.D. designed the study. A.I.T., A.G., N.T., H.M., P.C., C.C., and M.P.D. planned the fieldwork and logistics for Robert Island. A.I.T., A.G., H.M., and M.P.D. carried out all fieldwork (funding detailed in the acknowledgements). A.I.T., A.G., N.T., H.M., and M.P.D. carried out field sample processing. A.G. carried out all spectral and remote sensing analysis. A.I.T. carried out all molecular library preparation and phylogenetic analysis. A.I.T. led the writing of the manuscript with significant input from A.G., and all other co-authors contributing and editing the text. All authors have seen and approved the final version.

## Competing interests

The authors declare no competing interests.
