## [Peer Review file · Nature Communications]

Surface darkening by abundant and diverse algae on an Antarctic ice cap

Corresponding Author: Dr Matthew Davey

Version 0:

Reviewer comments:

Reviewer #1

(Remarks to the Author)

Thomson et al present an account of (glacier) algae on the Robert Island ice cap with particular attention to cellular abundance, species diversity and some IRF back-of-the envelope calculations. Altogether this is a nice piece of work that will be of high interest to the research community. It includes some novel phenotype and sequence data of Antarctic glacier algal "species" that is very interesting in particular. It is well written throughout. I have some specific comments on the manuscript itself to address below and generally I think the paper perhaps emphasises too much the novelty of glacier algal presence in Antarctica and degree of glacier algal impacts within their datasets (though it does a nice job here of providing some solid initial datasets) and I think the IRF focus is a good target to include but is more back of the envelope here than currently displayed.

Specific comments:

Abstract:

L20 - 10-13% by count/biomass/sequence data? Pretty low proportion.

L23 - what proportion of total melt is 70 cm MWE? And why is this expressed as snow melt rather than ice melt?

L45: poor wording at end of sentence "inhabiting the covering snow and ice..." - please reword.

L53: Ref 24 used here for 40 days or more bloom persistence is a "dark ice" paper and not a direct glacier algal bloom paper, i.e. they infer glacier algal presence from a harmonised albedo product, but do not measure actual presence.

L71: bio albedo reduction already should be BAR - check throughout.

L88: what is meant by "surface algae" - snow or glacier algae - looking at methods one retrieval algorithm has been used to map the algae here that was developed for snow algae (?) - how are the authors sure this is applicable to glacier algae (what ground truth data do they have?).

L96: It seems glacier algae are a small component of the community here, (max recorded density is 23% of the cells in that given sample) - but the tone of the paper (like in the title, abstract, etc) pushes the focus for things like IRF contributions just on to the glacier algae. Clearly, red snow algae, well studied in this general region, are also pulling their weight here, and the authors cannot distinguish the relative contributions of the two components to bio-albedo-reduction (Fig 5b tries this, but not sure how "Ancylnema dominated" samples were properly defined if the max Ancylnema count represented just 23% of the cells in its sample??).

Table 1 - Similarly, perhaps some more granularity in counts could be added here, 2.1×10^5 cells ml⁻¹ - it seems there are snow and ice samples included from the text above - why not add in at least this level of detail e.g. Ancylnema max density on ice and on snow?

Fig 2 - Were spectra and cell density samples paired throughout?

Fig 2 - Remote cell density estimates are for both communities (red snow and glacier algae), derived from a snow algal algorithm - again this clashes a little with the tone of the paper pushed toward glacier algae, when this is a mixed community assessment.

L107 - this is the really novel and interesting part of the paper.

L116 - not sure what is meant by "distribution of morphotype was limited to presence/absence data"? Can you expand on this for me please? Why do you not have cell counts of the different morphotypes?

Table 2 and Fig 3 - really like this info and imagery.

Fig 4 - Perhaps better rendered as an x/y scatter which may presumably distribute the cell-size-classes better and allow a more intuitive comparison across locations?

L152 - why is 92% similarity the threshold for ITS2 haplotypes - this seems very low.

Fig 6 - would benefit from a visual legend for the region colours displayed.

L200 - are there counts to support "non-bloom" statement - always worried of assumptions of lack of cell colonisation only due to macroscopic categorisation by observers.

L202 - are there published spectra available to aid this?

L209 - fine, but also glacier algae will quench reflectance across the entire visible range, could this also impact these trends?

L218 - what does "similar relationship" mean - please qualify this with some numbers

L22 - perhaps an assumption here that the other 59% of the variance is also due to (indirect) cell presence - what about purely physical changes in snow/ice structure?

L225 - can you justify why snow and not ice melt calculated here? Presumably the lower snow density will increase melt projections? How do your calculations play out for glacier ice densities? Perhaps a range should be reported here.

L227 - assuming a static bloom size over 122 day growth season, which is unrealistic. Not sure how far these back of envelope calculations can be meaningfully pushed?

L230: Did the authors calculate the % of melt during the study day this corresponds to? This context would be helpful.

L245: add in "Robert Island" before Antarctica.

L245:247: Again there is a subtle switch here from the true mixed community assessed here, to a total focus on glacier algae blooms - which was itself not quantified - just "surface algae" which included everything. Arguably glacier algae were a small component of your community. I think there needs to be a re-toning of how this is communicated throughout the paper.

L253 - very interesting results nicely discussed.

L285: Over emphasis of the potential role of glacier algae here not supported by the mixed community analysis.

Comparisons are made only to studies on glacier algal cell densities and impacts from Greenland - but those Greenland communities were 99% glacier algae dominated, whereas here glacier algae were max 23% of cell counts - thus the data are not truly comparable (or at least it should be made more clear what is being compared here).

L300 - change ice-algal to glacier algal.

L318 - "potential scale of glacier algal presence across Antarctica, and its ramifications" - as per earlier comments, I find this to be over-egging their relative importance in your datasets and the scope of the work.

L320 - similar comment for "significant biological contributions to glacier melt dynamics" - a 41% relationship between IRF and mixed cell counts is reported with no account of the proportion of total melt attributable to these communities (or glacier-algal-specific impacts).

Generally I think the conclusions misses the real novelty here - the unique glacier algal species/strains identified from the region.

L340 - how to sample ice with a tube?

L379 - same comment as before the 92% similarity.

L420 - "we separated samples based on the dominant taxa present within the spectrometers FoV"...what does this mean and how was it actually achieved? Presumably based on proper cell counts after spectra were obtained and the snow/ice surface was sampled? How can Ancyronema have dominated any FOV if their maximum proportional abundance (cell counts) was 23% as reported in the text?

L446 - 448: how are you able to use a snow algal product to estimate glacier algal cell abundance?

L458 - can the authors estimate the melt rates for their study location/day to contextualise the melt equivalent calculated?

Reviewer #2

(Remarks to the Author)

This work is the nucleus for unravelling the ecological significance of photosynthetic microbes living on Antarctic melting snow, and particularly, ice surfaces. While in the Northern hemisphere much is known about the role of cryoflora in terms of albedo change/melting acceleration, biodiversity or carbon sequestering, there is a huge gap for the circumstances at the southernmost continent on Earth, Antarctica. The sound results of this study will be a first milestone for a long-term project to investigate the whole Antarctic coastal regions, not only by satellite imagery but also performing field work wherever possible.

The manuscript is nice to read and most of the relevant literature was cited. Regarding details, I made comments and suggestions in the attached PDF.

Care has to be taken when talking about biogeographical conclusions. So far, it looks like as if Antarctica harbours most biodiversity in the genus Ancyronema (which makes sense), which is supported by the molecular data in this study. However, we still have sampling and sequencing gaps of the cryosphere in large parts of South America and in the Asian

Himalayas. Before this gap has not been filled, final conclusions about the evolution of glacier ice algae are not possible. Additionally, quite recently a mesophilic *Ancylonema* from moors has been found and described, a fact that should be considered in the discussion about the evolution of these zygnematophytes (a showcase that this genus is not obligatory on ice):

Busch, A., Slominski, E., Remias, D., Procházková, L., & Hess, S. (2024). A mesophilic relative of common glacier algae, *Ancylonema palustre* sp. nov., provides insights into the induction of vacuolar pigments in zygnematophytes. *Environmental Microbiology*, 26(8), e16680.

The dark phenolic pigmentation is well addressed; however I miss citing a detail of Ling & Seppelt when they reported a species "*Mesotaenium berggrenii*" with a rather ochre vacuole color! I found this organism myself many years ago at King George Island (unpublished), which is not far from the sampling site of the authors. I cannot find this striking species on the LM pics and so we can only speculate if this likely endemic, new *Ancylonema* is somewhere in the molecular data present or not. Please mention this reference and the difference in secondary pigmentation in the text.

This study is about surface darkening by diverse glacier algae, however the results almost exclusively address the true glacier ice algae. I suggest adding a few sentences about the biodiversity of true snow algae found (chlorophytes, chrysophytes), as they contribute much to the albedo change which their pigments. For the same reason, the title of the manuscript can be modified, e.g. to "... diverse snow and glacier ice algae on ...". Finally, the fact that other cryoflora than *Ancylonema* is very abundant and diverse in the South Shetland Islands can be cited by another reference, e.g.:

Soto, D. F., Gómez, I., & Huovinen, P. (2023). Antarctic snow algae: unraveling the processes underlying microbial community assembly during blooms formation. *Microbiome*, 11(1), 200. <https://doi.org/10.1186/s40168-023-01643-6>

Version 1:

Reviewer comments:

Reviewer #1

(Remarks to the Author)

The authors have done an excellent job in revising their manuscript based on both reviewers comments. They have addressed all of the comments from myself, re-focussing the paper on the wider diversity of algae they have found in their study region, whilst still nicely emphasising the interesting aspects associated with the *Ancylonema* community present. The updated melt impacts of the algal community seems more in-line with the scope of the work. This work remains of high interest to a wide readership and will foster many more interesting discoveries. This work is now ready for full acceptance in my opinion. Apologies for my slow re-review which was due to fieldwork absence. Best wishes, CW.

Reviewer #2

(Remarks to the Author)

The authors did an excellent job during revision!

Reviewers' comments and authors' responses to manuscript: Surface darkening by abundant and diverse algae on an Antarctic ice cap.

Thank you to both reviewers for their detailed and constructive comments. In particular, we are grateful for the feedback concerning the focus on *Ancylonema* without clearer consideration of the wider algal community, the feedback concerning overconfidence in our BAR and melt estimations, and the need for more methodological detail to support the findings. We have addressed all of these points, as well as the helpful error corrections and improved wording and figure suggestions, in the resubmitted manuscript. We have clarified our findings and conclusions to consider the composition, significance, and BAR impacts of the whole algal community on the Robert Island ice cap, as suggested by both reviewers, as supposed to just the *Ancylonema* group, which made up a relatively small proportion of the whole community. We then showcase diversity observed within *Ancylonema* as a striking example of the biodiversity and regional endemism present within Antarctic ice cap communities.

We have also addressed the comments around our initial BAR calculations and the contribution of *Ancylonema* within that, by considering the BAR effects of the community as a whole; by simplifying the satellite detection method to a binary algal presence-absence with a confidence threshold, and by reducing the extrapolation of the results to just consider the potential melt contributions to days with validated field measurements. We also discuss the limitations inherent in these analysis in more detail, including the difficulties of detecting and separating *Ancylonema* and other organic signals from the weathering crust.

We maintain that the presence and extent of algal blooms on the Robert Island ice cap represents a major terrestrial ecological system that has been overlooked in the Peninsula region and the wider Antarctic region, and conclude that although the BAR contributions are less than those previously reported from Greenland, these still represent an important contribution of biological communities to melt processes on the Robert Island ice cap. This factor should be considered in future modelling approaches. This communication will act as a primary reference point to address major questions around microbial photosynthetic cryosphere primary productivity, endemic biodiversity, and BAR contributions to melt in the Antarctic region.

Reviewer 1

Line	Comment	Response
n/a	Thomson et al present an account of (glacier) algae on the Robert Island ice cap with particular attention to cellular abundance, species diversity and some IRF back-of-the envelope calculations. Altogether this is a nice piece of work that will be of high interest to the research community. It includes some novel phenotype and sequence data of Antarctic glacier algal “species” that is very interesting in particular. It is well written throughout. I have some specific comments on the manuscript itself to address below and	Thank you for reviewing the manuscript and the positive comments, we are pleased that you found the paper very interesting. We have been through all the comments and provided a response and updates where required.
n/a	generally I think the paper perhaps emphasises too much the novelty of glacier algal presence in Antarctica and degree of glacier algal impacts within their datasets (though it does a nice job here of providing some solid initial datasets) and	We thank the reviewers for their emphasis of this point. This has made us re-appraise the data and reframe the focus of the results from Ancylonema contributions to broader ‘algal community’ contributions to BAR. We agree that this represents a more accurate appraisal of BAR contributions from algae on the ice cap, as well as a more accurate appraisal of the community and its significance in an ecological context.
n/a	I think the IRF focus is a good target to include but is more back of the envelope here than currently displayed.	We have updated our estimates of BAR contributions by assessing whole community impact on BAR, applying a more conservative presence absence method to algal bloom detection, and by simplifying our calculations of melt contribution to just consider total melt on days with validated field measurements. We hope that the updated results reflect a more confident, accurate and nuanced assessment of BAR impact on the ice cap.

Line	Comment	Response
L20	10-13% by count/biomass/sequence data? Pretty low proportion.	Thanks for pointing this out. We have removed this line as part of the reframed focus on total algal community on the ice cap.

L23	what proportion of total melt is 70 cm MWE? And why is this expressed as snow melt rather than ice melt?	We have reworked the melt estimates so that they are not based on the remote sensing data. We agree that there is likely not enough data and too many potential confounding factors to relate cell density to in-pixel chlorophyll absorption. We have replaced the initial estimate with an estimate based on a simple degree day model under two weather scenarios (from weather station data). Based on this the updated estimate of snow/weathering crust melt is around 0.5 to 2.3 % of total melt. We have updated the methods results, and discussion around this in in the text.
L45	poor wording at end of sentence “inhabiting the covering snow and ice...” - please reword.	Sentence rewritten for accuracy: “These communities are supported by photoautotrophic production from cyanobacteria and algae inhabiting surface and near-surface snow and ice”
L53	Ref 24 used here for 40 days or more bloom persistence is a “dark ice” paper and not a direct glacier algal bloom paper, i.e. they infer glacier algal presence from a harmonised albedo product, but do not measure actual presence.	This is a valid point, and we have updated the reference and observed growing season accordingly. The updated reference, Stibal et al. 2017, reported actively growing communities of zygnematophyte algae over a period of 35 days on the Greenland Ice Sheet.
L71	bio albedo reduction already should be BAR - check throughout.	corrected
L88	what is meant by “surface algae” - snow or glacier algae - looking at methods one retrieval algorithm has been used to map the algae here that was developed for snow algae (?) - how are the authors sure this is applicable to glacier algae (what ground truth data do they have?).	We have improved the wording to make this clearer: “Within the 13.8 km² area analysed from the 6th February 2023 WorldView 2 satellite image, algae covered an estimated 2.7 km², almost 20% of the snow, weathering crust and glacial ice shown in Fig. 2.” We have tested the retrieval method on our field spectra and we are confident that the method worked, including for Ancylonema containing samples. However, we have removed the cell density estimates since the cell relationship, as pointed out by the reviewers, was poor. We have added these details to the methodology.
L96	It seems glacier algae are a small component of the community here, (max recorded density is 23% of the cells in that given sample) - but the tone of the paper (like in the title,	We thank the reviewers for their emphasis of this point. This has made us re-appraise the data and reframe the focus of the results from Ancylonema contributions to broader ‘algal community’ contributions

	abstract, etc) pushes the focus for things like IRF contributions just on to the glacier algae. Clearly, red snow algae, well studied in this general region, are also pulling their weight here, and the authors cannot distinguish the relative contributions of the two components to bio-albedo-reduction (Fig 5b tries this, but not sure how “Ancydonema dominated” samples were properly defined if the max Ancydonema count represented just 23% of the cells in its sample??).	to BAR. We agree that this represents a more accurate appraisal of BAR contributions from algae on the ice cap, as well as a more accurate appraisal of the community and its significance in an ecological context. We have reframed the analysis and have made it clear that the algal forcing of albedo reduction and melt is based on the community spectra rather than on Ancydonema or snow algae separately.
Table 1	Similarly, perhaps some more granularity in counts could be added here, 2.1×10^5 cells ml⁻¹ - it seems there are snow and ice samples included from the text above - why not add in at least this level of detail e.g. Ancydonema max density on ice and on snow?	We have moved table 1 to supplementary data table S5 as the comparison of Ancydonema densities alone is not reflective of the ice cap algal community as a whole. However we feel the comparisons are still useful and valid. We feel it is notable that the cell counts on Robert Island are comparable in density to other Ancydonema dominated systems, despite Ancydonema in Robert Island communities being a much smaller proportion of the whole algal community. The variation in cell counts among habitat types – snow vs ice – is not as clear cut on Robert Island as you might find in other similar locations as the ice cap habitat is neither in typical snow nor ice, but a form of neve or firn. We have updated the text to be consistent in our description of the habitat observed which we describe as “neve-like weathering crust”, and have provided example imagery in supplementary figures S1a and S1b.
Fig 2	Were spectra and cell density samples paired throughout?	Yes, we have clarified this in the text by adding “and paired cell density” to the Figure caption.
Fig 2	Remote cell density estimates are for both communities (red snow and glacier algae), derived from a snow algal algorithm - again this clashes a little with the tone of the paper pushed toward glacier algae, when this is a mixed community assessment.	On re-assessment we agree we do not have enough data from the field spectra alone to separate Ancydonema and snow algae contributions and end up extrapolating too far. We have replaced this analysis with a binary algal cell presence/absence approach above a certain density threshold (details provided in methods) which accounts for whole

		algal community presence. In addition we have discussed the limitations of separating organic and inorganic contributions, as well as the difficulties of separating Ancylonema signal in more detail in the discussion.
L107	this is the really novel and interesting part of the paper.	Thank you, we appreciate the comment.
L116	not sure what is meant by “distribution of morphotype was limited to presence/absence data”? Can you expand on this for me please? Why do you not have cell counts of the different morphotypes?	Morphotype data was taken from Dinolite microscopy imagery – which was not quantitative. Quantitative cell count data was taken using the Countess Cell Counter – which did not allow clear differentiation of different Ancylonema morphotypes (but did allow ID of Ancylonema through distinctive chloroplast shape and organisation under autofluorescence). The data presented in figure S2 were compiled from presence/absence data from the Dinolite imagery. But we agree that this is not informative without more qualitative data, and have removed figure S2 and the discussion around Ancylonema morphotype distributions. We intend to follow up this manuscript with a more detailed study looking at community composition and distribution across Robert Island. This will look in more detail at differences in distribution amongst Ancylonema morphotypes, as well as wider cryophilic algal community members.
Table 2 and Fig 3	really like this info and imagery.	Thank you.
Fig 4	Perhaps better rendered as an x/y scatter which may presumably distribute the cell-size-classes better and allow a more intuitive comparison across locations?	Thankyou for the suggestion. We have updated Figure 4 to include an xy scatter of length vs width across all Ancylonema cell types from Robert Island. This nicely highlights the cell width difference between morphotypes RI-N and the other two large morphotypes RI-E and RI-T. We feel the literature comparisons provided by the distribution plots are a useful inclusion, and compliment the xy y scatter plot, and as such we have kept these in the figure. We have also updated the colour scheme of the distribution plots to match the scatter plot, and

		removed the colour from the literature comparison datapoints to avoid confusion. We have updated the caption accordingly.
L152	why is 92% similarity the threshold for ITS2 haplotypes - this seems very low.	The rRNA ITS2 region is considered a hypervariable marker region, which confers a greater phylogenetic resolution amongst closely related groups. The short nature of the metabarcode 'mini-barcode', and the hypervariable nature of the marker, mean a greater dissimilarity is expected amongst related species. This threshold represents the greatest distance observed between our reference sequences and a putative Ancylonema asv in our datasets. As a comparison, Remias et al. 2023 apply an 89% similarity threshold to identify closely related ITS2 sequences from their asv dataset.
Fig 6	would benefit from a visual legend for the region colours displayed.	We have updated Figure 6 to include a visual legend for regions
L200	are there counts to support "non-bloom" statement - always worried of assumptions of lack of cell colonisation only due to macroscopic categorisation by observers.	Yes these were spectra of snow and weathering crust where no cells were detected under fluorescence using the automated cell counter. We have added this clarification in the text.
L202	are there published spectra available to aid this?	Not pure reflectance spectra, no.
L209	fine, but also glacier algae will quench reflectance across the entire visible range, could this also impact these trends?	Yes, though I'm not sure that we have the data to quantify this properly. Added "the darker pigmentation of Ancylonema cells is also likely to influence this"
L218	what does "similar relationship" mean - please qualify this with some numbers	We have removed the statement and updated the paragraph to reflect the whole algal community cell relationship.
L220	perhaps an assumption here that the other 59% of the variance is also due to (indirect) cell presence - what about purely physical changes in snow/ice structure?	We agree with this point and have added the clarification below: "explained only 41% of variance within the data, indicating a significant contribution to IRF from secondary effects of cell presence, such as enhanced liquid water and mineral entrainment as well as differences in snow and ice structure."
L225	can you justify why snow and not ice melt calculated here? Presumably the lower snow density will increase melt	We have tested this relationship temperature, irradiance, broadband reflectance and snow and ice density measurements from the field.

	projections? How do your calculations play out for glacier ice densities? Perhaps a range should be reported here.	The results are reported in Table 2. There was very little difference in measured density between the weathering crust and snow on the ice cap. The snow on the ice cap was notably very saturated. The recalculated melt amounts are now less since we have included passive albedo effects in the calculations.
L227	assuming a static bloom size over 122 day growth season, which is unrealistic. Not sure how far these back of envelope calculations can be meaningfully pushed?	Agreed, we have removed these extrapolations.
L230	Did the authors calculate the % of melt during the study day this corresponds to? This context would be helpful.	We have calculated this and have incorporated them into the results, Table 2, and the discussion.
L245:247	Again there is a subtle switch here from the true mixed community assessed here, to a total focus on glacier algae blooms - which was itself not quantified - just “surface algae” which included everything. Arguably glacier algae were a small component of your community. I think there needs to be a re-toning of how this is communicated throughout the paper.	We have adjusted the text throughout the manuscript to reframe the work around the community composition, diversity and BAR effects of the total algal community on the Robert Island ice cap, as opposed to just glacier algae. We agree that this is a more accurate appraisal of the ecosystem and its significance. We maintain that the presence and scale of the total algal community on the ice cap is significant in the context of Antarctic terrestrial productivity.
L253	very interesting results nicely discussed.	Thank you
L285	Over emphasis of the potential role of glacier algae here not supported by the mixed community analysis. Comparisons are made only to studies on glacier algal cell densities and impacts from Greenland - but those Greenland communities were 99% glacier algae dominated, whereas here glacier algae were max 23% of cell counts - thus the data are not truly comparable (or at least it should be made more clear what is being compared here).	We have updated the manuscript to reflect the fact that the BAR calculations made are for mixed algal communities, not just Ancylonema . We have tried to make it clearer that the comparison to the Greenland ice sheet is comparing two different biological systems, one dominated by Ancylonema and one of mixed assemblage. And that this in itself is interesting. It is difficult to find any other examples of BAR calculations from ice sheet/glacier surfaces. Other examples of BAR calculations, for instance from Khan et al, Engstrom et al, and others are based on red snow algae and seasonal snow or supra-glacial snow habitats.

		We have followed this section with a critical summary of the limitations and conclude that section with a statement that more examples from glacier surfaces are needed to better constrain glacier surface BAR estimations.
L300	change ice-algal to glacier algal.	Corrected
L318	“potential scale of glacier algal presence across Antarctica, and its ramifications” - as per earlier comments, I find this to be over-egging their relative importance in your datasets and the scope of the work.	We have updated this statement to consider all algae on the ice cap, rather than just Ancylonema , But we are confident that the densities of algal blooms, and the extent observed in our study area are significant in an Antarctic context. The current best estimates for terrestrial photosynthetic aerial coverage from Walshaw et al 2024 for instance were not able to include red snow algae in their accounting due to the detection methods used. The fact that the area studied here equates to 6% of that estimate alone suggests that ice cap algal blooms across the South Shetlands and the Antarctic peninsula more widely may make up a huge proportion of the aerial coverage of photosynthetic ecosystems in terrestrial Antarctica. What this equates to in terms of primary productivity is another major question that will require answering.
L320	similar comment for “significant biological contributions to glacier melt dynamics” - a 41% relationship between IRF and mixed cell counts is reported with no account of the proportion of total melt attributable to these communities (or glacier-algal-specific impacts).	We have updated the results and discussion and included the estimates in table 2 to provide a more conservative and nuanced analysis of the BAR contributions of algae on the Robert Island ice cap. We maintain that the updated estimates (0.5 - 2.3% of total melt based on conditions from the days measured in table 2), though much less than those observed in Greenland, still represent a significant contribution of BAR to melt processes on the ice cap, and one that should be taken into consideration in future mass balance modelling efforts for Antarctic glaciers and ice sheets.
L320	Generally I think the conclusions misses the real novelty here - the unique glacier algal species/strains identified from the region.	Whilst we agree that the diversity in Ancylonema is a significant finding within the study, we feel that the extent and contribution of algae on ice cap habitats to photoautotrophic ecosystems in the region, and to terrestrial ecology of Antarctica more widely, may be the most significant aspect of the study (see above comment on Walshaw et al.

		2024 estimates). We feel that the BAR calculations are also critical to highlight in order to help advance the discussion around ice retreat and changing landscapes in the region and incorporate these factors into future modelling approaches.
L340	how to sample ice with a tube?	We have updated the wording to “taken from the top 3 cm of snow or weathering crust”. The habitat across most of the ice cap surface consisted of neve-like weathering crust ‘pillows’ divided by drainage water channels. The neve weathering crust was composed of coarse, firm snow-ice crystals. These were penetrable with a sampling tube, as supposed to the hard ice substrate described from other glacier sites. Sampling tubes were filled by pushing the tube along the surface, at a depth of ~3cm, filling the tube to the top (50 ml or 15 ml) whilst avoiding compaction. We agree that this can lead to sampling/volume inaccuracies, but we do not feel these will majorly affect the results presented here. We look forward to the development of improved consensus methods for snow and ice sampling to address some of these concerns.
L379	same comment as before the 92% similarity.	Please see the above response.
L420	“we separated samples based on the dominant taxa present within the spectrometers FoV”...what does this mean and how was it actually achieved? Presumably based on proper cell counts after spectra were obtained and the snow/ice surface was sampled? How can Ancylonema have dominated any FOV if their maximum proportional abundance (cell counts) was 23% as reported in the text?	We agree with this point and have re-assessed this approach and removed the analysis based on the categorisation of “ Ancylonema dominated” samples. Instead we have compared samples with Ancylonema present or not present, based on cell counts (Countess images) and community analysis (LM microscopy).
L446 - 448	how are you able to use a snow algal product to estimate glacier algal cell abundance?	Please see the above response.
L458	can the authors estimate the melt rates for their study location/day to contextualise the melt equivalent calculated?	The estimates are now provided in Table 2.

Reviewer 2

Line	Comment	Response
n/a	This work is the nucleus for unravelling the ecological significance of photosynthetic microbes living on Antarctic melting snow, and particularly, ice surfaces. While in the Northern hemisphere much is known about the role of cryoflora in terms of albedo change/melting acceleration, biodiversity or carbon sequestering, there is a huge gap for the circumstances at the southernmost continent on Earth, Antarctica. The sound results of this study will be a first milestone for a long-term project to investigate the whole Antarctic coastal regions, not only by satellite imagery but also performing field work wherever possible. The manuscript is nice to read and most of the relevant literature was cited. Regarding details, I made comments and suggestions in the attached PDF.	Thank you for reviewing the manuscript and we appreciate the positive review. We have taken into account the comments and have made edits where required to clarify the points and provide extra information.
n/a	Care has to be taken when talking about biogeographical conclusions. So far, it looks like as if Antarctica harbours most biodiversity in the genus Ancylonema (which makes sense), which is supported by the molecular data in this study. However, we still have sampling and sequencing gaps of the cryosphere in large parts of South America and in the Asian Himalayas. Before this gap has not been filled, final conclusions about the evolution of glacier ice algae are not possible.	We have highlighted the pattern of high diversity and regional endemism in Ancylonema from Robert Island, and allude to the fact that the patterns of morphological and phylogenetic diversity seen would be consistent with a hypothesis of refugial expansion from Antarctica in this lineage of Ancylonema. We are aware that there are major gaps in the global analysis of glacial Ancylonema, and that a more thorough global dataset will be required to more firmly support these hypothesis. However we feel that our findings do support these lines of enquiry, and it would be remiss not raise these questions in the discussion. We also feel that the diversity and regional endemism

		observed in Ancylonema is a possible indicator of regional endemism in other cryoflora groups present in the Robert Island community. We point this out in the reframed discussion, using the Ancylonema investigation as a showcase for these processes of microbial biogeography and regional-endemism in Antarctic cryoflora communities
n/a	Additionally, quite recently a mesophilic Ancylonema from moors has been found and described, a fact that should be considered in the discussion about the evolution of these zygnematophytes (a showcase that this genus is not obligatory on ice): Busch, A., Slominski, E., Remias, D., Procházková, L., & Hess, S. (2024). A mesophilic relative of common glacier algae, Ancylonema palustre sp. nov., provides insights into the induction of vacuolar pigments in zygnematophytes. Environmental Microbiology, 26(8), e16680.	The recent paper by Busch et al. 2024 is very interesting, and we are grateful to be able to include and address the findings in our manuscript revisions. We have updated the 18S phylogeny in figure S4 accordingly. We have also updated the text in results to acknowledge this update to the Ancylonema genus. From a biogeographic and evolutionary point of view, we feel this is beyond the scope of this manuscript, which focusses on the diversity within glacial lineages of Ancylonema as an example of cryoflora regional endemism in Antarctica. From what evidence is available, these lineages appear to have diverged from the Ancylonema palustre lineage some time before their radiation amongst the ice-dependant species and lineages seen today. The question of when that divergence occurred should be a high priority in order to constrain a timeline of when these glacial lineages may have first colonised ice habitats. However, this represents a somewhat separate question to the question of whether glacial Ancylonema lineages have undergone a process of refugia and expansion since then. We look forward to seeing more research in this area, including more rigorous testing of refugial hypotheses, as well as estimation of divergence ages amongst mesophilic and psychrophilic lineages of Ancylonema, which might help close the gap between contemporary patterns of biogeography, recent expansion histories, and deeper evolutionary histories, as laid out by Bowles et al. 2024.

n/a	The dark phenolic pigmentation is well addressed; however I miss citing a detail of Ling & Seppelt when they reported a species "Mesotaenium berggrenii" with a rather ochre vacuole color! I found this organism myself many years ago at King George Island (unpublished), which is not far from the sampling site of the authors. I cannot find this striking species on the LM pics and so we can only speculate if this likely endemic, new Ancydonema is somewhere in the molecular data present or not. Please mention this reference and the difference in secondary pigmentation in the text.	Whilst we did not see the reddish-brown coloration in fresh samples, we did see a more ochre hue in cells that had undergone freezing and transport (as shown in Figure S3). We have made a note of the distinction in colour in Table 1. We also hypothesize that the strong sunlight exposure at the ice cap surface on Robert Island likely plays a role in intensifying darker pigmentation in these cells and this factor might account for some of the colour variability. It is also worth noting that Ling & Seppelt used a Lugol's iodine stain prior to observation, which could enhance the brownish ochre coloration and potentially affect darker pigments (as seen with formalin treatment or freezing). Nevertheless, we share your enthusiasm that such differences might mean the possibility of more Ancydonema species in Antarctica!
n/a	This study is about surface darkening by diverse glacier algae, however the results almost exclusively address the true glacier ice algae. I suggest adding a few sentences about the biodiversity of true snow algae found (chlorophytes, chrysophytes), as they contribute much to the albedo change which their pigments.	Thank you for raising this. After considering this point, as well as similar points raised by reviewer 1 we have reframed the study to consider the whole algal community in our assessment. We agree that this is a more accurate appraisal of the community and its impacts considering that Ancydonema make up a small proportion of the total community. We have provided a more detailed results section as well as discussion on the community composition based on LM microscopy assessment as well as from the 18S metabarcode dataset. We hope this is an interesting and relevant addition to the study.
n/a	For the same reason, the title of the manuscript can be modified, e.g. to "... diverse snow and glacier ice algae on ...".	In line with the above, we have updated the title of the manuscript to: "Surface darkening by abundant and diverse algae on an Antarctic ice cap".
n/a	Finally, the fact that other cryoflora than Ancydonema is very abundant and diverse in the South Shetland Islands can be cited by another reference, e.g.:	Thank you, we are aware of this publication and have now added it to the manuscript.

	Soto, D. F., Gómez, I., & Huovinen, P. (2023). Antarctic snow algae: unraveling the processes underlying microbial community assembly during blooms formation. Microbiome , 11(1), 200. https://doi.org/10.1186/s40168-023-01643-6	
--	--	--

Line	Comment	Response
Title	Reviewer: Consider title update, as snow and glacier ice algae contribute.	Title updated to “Surface darkening by abundant and diverse algae on an Antarctic ice cap” to reflect the wider algal community present on the Robert Island ice cap
L33	Meaning of ‘localised’?	Removed “localised” for clarity
L44	Change order to “algae, fungi and ...”	completed
L46	No capitalisation of zygnematophyte	corrected
L51	Change to “a purpurogallin derivative”	We have modified this section as part of the wider refocus on whole algal community. The sentence on Ancylonema pigments now reads “energy-absorbing dark purple phenolic pigments in their cells”. We feel this is an appropriate level of detail to understand the manuscript. Additional references regarding the pigment are provided.
L60	insert "ice" after “glacier”	Changed in line with refocus to “To date, algae blooms on glaciers”
L110	“as well as” – rephrase/wording	Sentence rephrased for clarity
Table 2	“RI-T” - Comment of reviewer: I saw this cell type also on tropical glaciers in Colombia, South America.	This is interesting! With this observation, and the likely observation of RI-E by N. Takeuchi in Chile, there is perhaps an interesting biogeographic pattern emerging in Southern hemisphere Ancylonema ?
Table 2	Correct M. berggrenii (two g’s)	Corrected, thanks
Table 2	“RI-C” – length and width numbers confused?	Many thanks for spotting this, corrected
Figure 3h	h) is not in focus and too dark - I cannot judge if this is var. "chodatii"	We apologise for the lower resolution of some of the micrographs in figure 3. The micrographs were taken in the field, using a DinoLite portable microscope. This is not a high resolution imaging system, and

		was susceptible to vibrations in the field station during inclement weather. However, we feel that the field images are a more accurate representation of the Ancylonema morphotypes observed in the field. We observed a significant loss of pigmentation in Ancylonema cells that had been frozen (which we describe in the text). We have changed the micrograph in panel h to give a better representation of Ancylonema var. RI-C, and have brightened the image. We have also removed the comparison to Ancylonema nordenskiöldii var. chodatti. The Robert Island form was more commonly in pairs or single, rather than filamentous, as described in Procházková et al. 2021. A more accurate comparison would be to the large M. berggreni morphotype observed by Ling and Seppelt (1990). We have adjusted the descriptions and references in table 1 accordingly, and corrected the terminology around the use of ‘morphotype’ in table 1 and in the caption for figure 3.
Figure 4	Meaning of colors not stated?	We have updated the colour scheme of the distribution plots to match the new xy scatter plot (suggested by reviewer 1), and removed the colour from the literature comparison datapoints to avoid confusion. We have updated the caption accordingly.
Figure 4	General comment: Best numbers for length include information about cell age, as they grow significantly after cleavage.	This is a valid point. The data presented for Robert Island populations was taken across the duration of the sampling season (7th January – 18th February 2023), and as such some of the large variability in length may be due to changes in the average cell age across the population in different samples and over time. Measurements taken during the division of Ancylonema cells may also lead to overestimates of ‘average’ single cell size. However we feel that the sample number (n = 230 for RI variants in fig. 4a), and the coverage of sampling across the seasons adequately captures the size range of the observed Ancylonema variants from Robert Island, and supports the morphological divergence from northern hemisphere populations (which will have been subject to similar variability in length over season and cell age).

L164	The CBC concept was not introduced earlier in the manuscript, the reader needs some more information or reference.	We have updated this sentence to include an explanation of sequence-structure alignment and the CBC concept, and have added references (Keller et al. 2008; Coleman, 2003; Muller et al. 2007) to support this and direct the reader to a more detailed explanation. A more detailed explanation is also included in the methods section.
Figure 6	Please state the origin of the sequence called "Remias_2023 H2" or rename. Tell accession number, was it personal communication?	This information was taken from Remias et al. 2023 supplementary figure 5 and supplementary table 5. Haplotype H2 (asv_455e90e031ab0918cb4d186912312d4c) is stated as "identical with the reference sequence WP211 – OL898470". The caption text has been updated to clarify this source.
L246	Insert "Maritime" after "in" and before "Antarctica"	completed
L247	References 48 (Khan et al. 2021) & 49 (Davey et al. 2019) are in this context not sufficient. There are further references for these local communities, e.g. Soto et al.	We have updated the references to include the recent work by Soto et al. 2023. Whilst we are aware of other studies that have characterised snow algal communities in Antarctica, using both traditional (Ling and Seppelt – Snow algae of the Windmill Isles), and molecular approaches (Luo et al. 2020, Camara et al. 2023), we do not feel these support the statement as well regarding the scale and extent of coastal snow algal bloom ecosystems.
L273	I guess for most readers it is not clear here what is the third reported Ancylonema species from ice?	Updated the text to "presence of the currently reported species" to avoid confusion around the three species. We feel it is clear in the results text that one of our haplotypes aligns with the putative Ancylonema sp. reported by Remias et al., and this supports this later statement in the discussion.
L277	"regional" endemism?	Corrected to 'regional-endemic' for clarity
L279	In Ancylonema "living on ice"	Amended to "cryophilic Ancylonema " for clarity and to differentiate from the mesophilic A. palustre
L291	what about stating it this way "on Sermersuaq (the Greenland Ice Cap)"?	Thank you for the suggestion. We have introduced the nomenclature in the introduction "the West Greenland Sermersuaq (Ice Sheet)," and have decided to keep "the Greenland Semersuaq" following this first instance. We feel the use of the Greenlandic name is most appropriate and hope that our initial clarification is understandable to the reader.
L303	State for which region was 5750 km ²	This sentence has been removed as part of the wider reframing.

L311	... and their overall photosynthetic productivity (should be mentioned somewhere here)	We have added “and evaluate their contribution to regional terrestrial productivity.” to highlight the importance of understanding contributions to terrestrial primary productivity.
L324	insert "blooms" after 'algae'	This sentence has been removed as part of the wider reframing
Fig S5	Add “secondary” structure to “ITS2 sequence structure”	corrected